# Long-Term Monitoring of Cloud Water Chemistry at Whiteface Mountain: The Emergence of a New Chemical Regime

Christopher E. Lawrence[1], Paul Casson[1], Richard Brandt[1], James J. Schwab[1], James E. Dukett[2], Phil Snyder[2], Elizabeth Yerger[3], Daniel Kelting[3], Trevor C. VandenBoer[4], and Sara Lance[1]

[1]Atmospheric Sciences Research Center (ASRC), University at Albany, SUNY ETEC building, 1220 Washington Ave, Albany NY 12226
[2]Adirondack Lake Survey Corporation (ALSC), 1115 NYS Rt.86, P.O. BOX 296, Ray Brook NY 12977
[3]Paul Smith's College Adirondack Watershed Institute (AWI), P. O. Box 265, Routes 86 and 30, Paul Smiths NY 12970
[4]Department of Chemistry, York University, Toronto, Ontario, Canada

**Correspondence:** Sara Lance (smlance@albany.edu)

**Abstract.**

Atmospheric aqueous chemistry can have profound effects on our environment. The importance of chemistry within the atmospheric aqueous phase started gaining widespread attention in the 1970s as there was growing concern over the negative impacts on ecosystem health from acid deposition. Research at mountaintop observatories including Whiteface Mountain (WFM) showed that gas phase sulfur dioxide emissions react in cloud droplets to form sulfuric acid, which also impacted air quality by increasing aerosol mass loadings. The current study updates the long-term trends in cloud water composition at WFM for the period 1994-2021 with special consideration given to samples that have traditionally been excluded from analysis due to inorganic charge imbalance. We emphasize three major findings: 1) a growing abundance of total organic carbon (TOC), with annual median concentrations more than doubling since measurements began in 2009, 2) a growing imbalance between the measured inorganic cations and anions, consistent with independent rain water observations, implying that a substantial fraction of anions are no longer being measured with the historical suite of measurements, and 3) a growing number of samples exhibiting greater ammonium concentrations than sulfate plus nitrate concentrations, which now routinely describes over one-third of samples. Organic acids are identified as the most likely candidates for the missing anions, since the measured inorganic ion imbalance correlates strongly with measured TOC concentrations. An "inferred cloud droplet pH" is introduced to estimate the pH of the vast majority of cloud droplets as they reside in the atmosphere using a simple method to account for the expected mixing state of calcium and magnesium containing particles. While the inferred cloud droplet pH closely matches the measured bulk cloud water pH during the early years of the cloud water monitoring program, a growing discrepancy is found over the latter half of the record. We interpret these observations as indicating a growing fraction of cloud droplet acidity that is no longer accounted for by the measured sulfate, nitrate and ammonium concentrations. Altogether, these observations indicate that the chemical system at WFM has shifted away from a system dominated by sulfate to a system controlled by base cations, reactive nitrogen species and organic compounds. Further research is required to understand the effects on air quality, climate, and ecosystem health.

# 1 Introduction

Whiteface Mountain (WFM) has been an important site for cloud water chemistry measurements dating back to the 1980s (Falconer and Falconer, 1980; Mohnen and Vong, 1993), with routine summertime cloud water monitoring beginning in 1994. Through field experiments at WFM and other locations in the Eastern U.S., researchers established that gas phase sulfur dioxide ($SO_2$), originating largely from fossil fuel combustion, can dissolve in cloud droplets and undergo aqueous oxidation to form sulfate ($SO_4^{2-}$). This chemistry has contributed substantially to particulate mass, posing significant health risks and contributing to cooling effects on the climate (Kiehl and Briegleb, 1993; Pope and Dockery, 2006; Myhre et al., 2013). Sulfate and nitrate ($NO_3^-$) are also the major causes behind acid deposition, a process that can have detrimental ecosystem effects such as decreased aquatic biodiversity, calcium depletion from soils, release of toxic aluminum (Gorham, 1998; Likens et al., 1998; Driscoll et al., 2001; Menz and Seip, 2004) and deterioration of building materials (Xie et al., 2004; Bravo A. et al., 2006). These growing societal and environmental problems helped influence and shape amendments to the Clean Air Act in the 1990s focusing on the criteria air pollutants $SO_2$ and $NO_x$. Starting in 1994, the U.S. Environmental Protection Agency (EPA) formed a network of the mountaintop sites in the Eastern U.S. through the Mountain Acid Deposition Program (MADPro) to begin monitoring the progress of Clean Air Act Amendments (Baumgardner et al., 2003). Regulation of these emissions proved to be successful, as $SO_2$ and to a lesser extent $NO_x$ have decreased for several decades in the U.S. (US EPA, 2019), leading to decreases in acid rain throughout the Eastern U.S., including New York State (Rattigan et al., 2017). Substantial decreases were also observed in cloud water acidity at WFM (Dukett et al., 2011), corresponding to decreases in particulate matter concentrations of $SO_4^{2-}$ and $NO_3^-$ (Schwab et al., 2016b). Many ecosystems in the Adirondacks previously impacted by acid deposition have shown signs of recovery (Driscoll et al., 2016).

Routine long-term measurements of cloud water or fog water are not widely available, but other locations worldwide have seen similar long-term reductions in $SO_4^{2-}$ and/or $NO_3^-$ concentrations with concomitant decrease in acidity, including California's central valley (Herckes et al., 2015), Pennsylvania (Straub, 2017), New Hampshire (Murray et al., 2013), the UK (Cape et al., 2015), Italy (Giulianelli et al., 2014), France (Deguillaume et al., 2014) and Japan (Yamaguchi et al., 2015; Watanabe et al., 2022). A thorough review of both long-term and short-term cloud and fog chemistry studies has been recently compiled by Isil et al. (2022).

The significant reductions in $SO_4^{2-}$ and $NO_3^-$, and therefore cloud water acidity, at WFM indicate a dramatically changed chemical system, with a growing contribution from less well-understood and more labile analytes such as organic compounds and ammonium ($NH_4^+$). Organic compounds are found ubiquitously throughout the atmosphere, playing key roles in ozone formation and aerosol physicochemical properties such as hygroscopicity, optical properties and reactivity (Jimenez et al., 2009). During a two-week pilot study at WFM in 2017, organic matter was shown to comprise as much as 93%, and an average of 78%, of submicron aerosol mass (Zhang et al., 2019). Similarly, total organic carbon (TOC), the sum of soluble and insoluble OC, is a major component within cloud and fog droplets (Herckes et al., 2013). Organic compounds are chemically complex, containing thousands of compounds, all with different chemical and physical properties spanning orders of magnitude in volatility and solubility, which dictates the chemical pathways they can take part in (Jimenez et al., 2009), making their

identification and potential environmental effects challenging to determine (Thornton et al., 2020). The controlling factors for TOC concentrations in cloud droplets are highly uncertain (Ervens, 2015). Organic compounds can also interact with inorganic ions within cloud droplets to form compounds like organic salts, organic nitrogen, and organic sulfur compounds (McNeill, 2015).

Reactive nitrogen deposition, from both reduced and oxidized forms, has also been gaining increased attention (Kanakidou et al., 2016; Stevens et al., 2018). While nitrogen can act as a nutrient in ecosystems that are nitrogen limited, excess deposition can lead to harmful environmental effects such as soil acidification by leaching out buffering base cations like calcium ($Ca^{2+}$), magnesium ($Mg^{2+}$) and potassium ($K^+$), and through nitrification, potentially mobilizing toxic metals such as aluminum (Lawrence and David, 1997; Tian and Niu, 2015). Nitrogen deposition, when combined with other factors, can also contribute to harmful algal blooms, which threaten aquatic ecosystems and human health (Paerl and Otten, 2013). A significant fraction of total nitrogen (TN) in cloud and rain water is organic nitrogen (ON), typically comprising ∼30% of water soluble nitrogen (Cape et al., 2011). Much like organic carbon, the sources of organic nitrogen, as well as the chemical aging, deposition, and ecosystem effects, are highly uncertain and require more research. Ammonia ($NH_3$) is the primary atmospheric base in Earth's atmosphere, and an unregulated gas phase pollutant with increasingly recognized importance to atmospheric chemistry. Despite emissions of $NH_3$ in the U.S. largely decreasing prior to 2015 (US EPA, 2019), since then atmospheric concentrations of $NH_3$ appear to be increasing (Warner et al., 2017; Liu et al., 2019; Yu et al., 2018). At the same time, across New York state, aerosol $NH_4^+$ concentrations have steadily decreased and wet precipitation $NH_4^+$ concentrations have remained relatively constant (Rattigan et al., 2017), suggesting that complex long-term shifts in $NH_3$ partitioning are taking place.

This paper provides an updated review of trends in summertime cloud water composition at WFM using data obtained from MADPro (1994-2000), the long-term monitoring conducted by the Adirondack Lake Survey Corporation (ALSC) (2001-2017) and recent measurements conducted under the oversight of the Lance Lab at the Atmospheric Sciences Research Center (ASRC) (2018-2021). While ASRC conducted the cloud water sampling in recent years, the majority of the chemical analysis was conducted by the Adirondack Watershed Institute (AWI) in 2018 and 2019 and by ALSC in 2020 and 2021. A detailed review of the sample collection methods and laboratory analyses over this period are also discussed, with a focus on major system changes that have not been fully described in previously published scientific papers. Previous works have reported trends in bulk cloud water composition at WFM up to 2013, largely focusing on analytes most relevant for acid deposition (Aleksic et al., 2009; Dukett et al., 2011; Schwab et al., 2016b). Here, a critical review of past data analysis methodologies is presented, alongside a thorough assessment and justification for modifications to these methodologies, which have resulted in significant changes to the long-term trends. In this paper, we also report the first long-term trends in cloud water TOC at WFM.

## 2 Collection History, Methods, Laboratory Procedure, and Data Analysis

### 2.1 Site Description and Collection Methods

Whiteface Mountain (WFM) is located in the "High Peaks" Region of the Adirondacks Mountains in Upstate, NY, with summit elevation of 1483m. The WFM site history is described in detail by Schwab et al. (2016a, b). It should be noted that the first

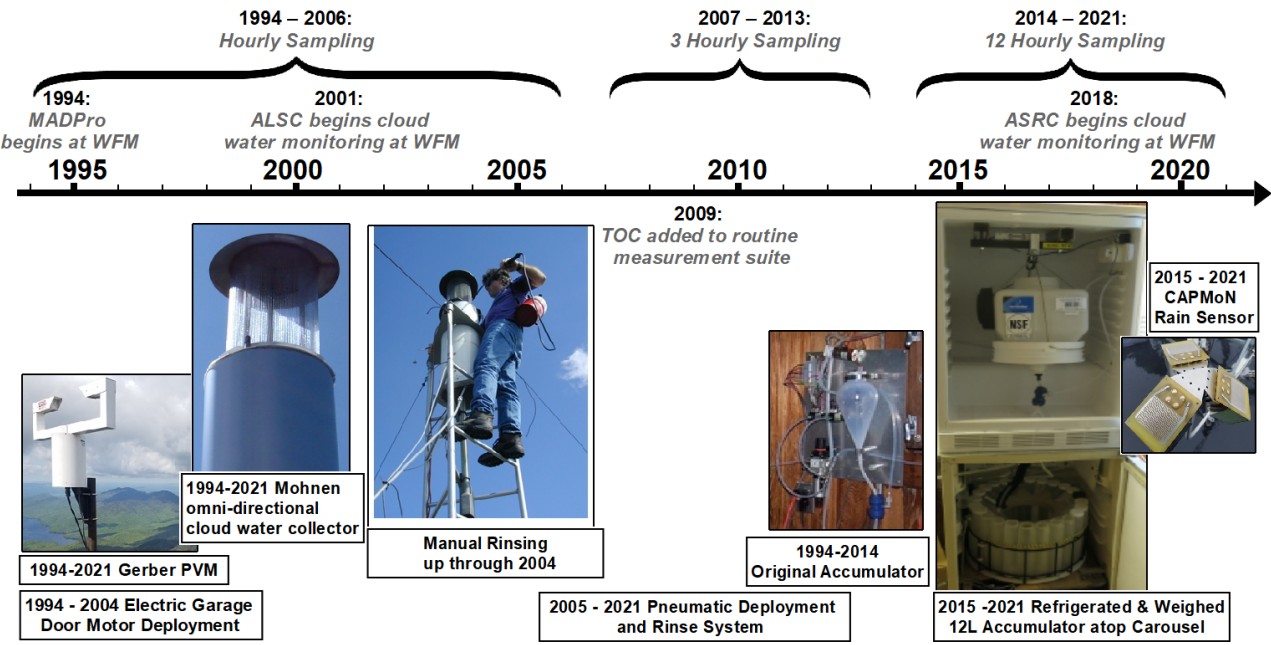

**Figure 1.** Timeline of major events in cloud water collection at WFM for the routine long-term data set analyzed in this paper (1994-2021), with sampling conducted by the Mountain Acid Deposition Program (MADPro) 1994-2000, the Adirondack Lake Survey Corporation (ALSC) 2001-2017, and the Atmospheric Sciences Research Center (ASRC) 2018-2021, and two major system overhauls in 2005 and in 2015. While ASRC conducted the cloud water sampling in recent years, the majority of the chemical analysis was conducted by the Adirondack Watershed Institute (AWI) in 2018 and 2019 and by ALSC in 2020 and 2021. The same basic protocols, cloud water collector and liquid water content sensor were used throughout this entire measurement period, but sampling frequency decreased from hourly to 3-hourly to 12-hourly, necessitating a larger accumulator vessel and an additional refrigerator to house it. Automated filtering began in 2018. Photos documenting the manual rinsing process and the original accumulator were obtained from Eric Hebert, Environmental Engineering and Measurement Services, Inc.

cloud water TOC measurements at WFM were conducted for a handful of cloud events in 1987, along with the first published organic acid measurements in cloud water at WFM. This powerful seminal study showed that potential sources of organic carbon including organic acids could share similar sources with $SO_4^{2-}$ and $NO_3^-$ (Khwaja et al., 1995). Water Soluble Organic Carbon (WSOC) measurements were also previously obtained for a selection of cloud water samples at WFM from 1990 to 1992 (Anastasio et al., 1994).

All of the bulk cloud water data reported in this paper used a Mohnen omni-directional passive cloud water collector (Mohnen, 1980)(Fig. 1). The collector contains two disks with 0.035-0.04mm Teflon strings strung vertically between the disks. As cloud droplets pass through the collector and collide with the Teflon strings, water beads up and eventually drips down the strings. The cloud water is then funneled by gravity through a tube and fed into a high density polyethylene vessel

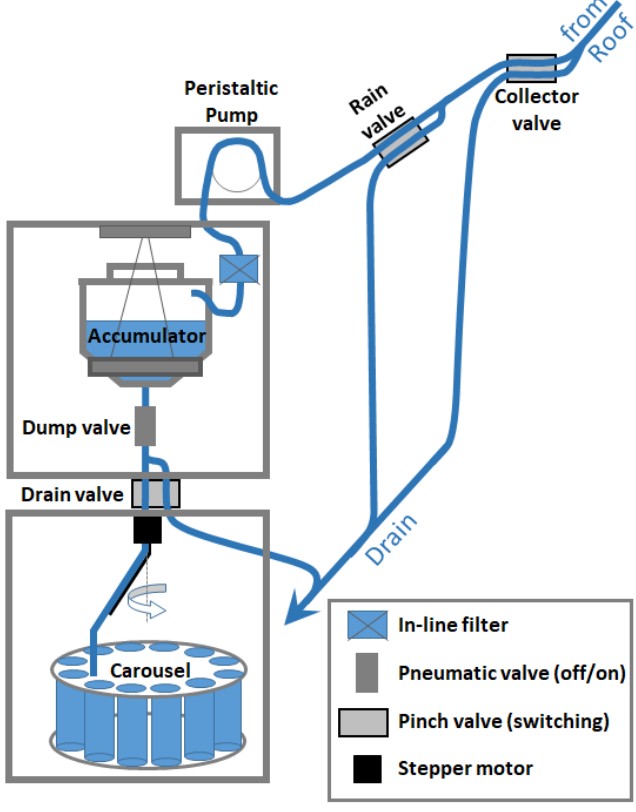

**Figure 2.** Schematic of the physical components of the current cloud water collection system inside the research observatory at the summit of WFM. A series of valves control where the collected cloud water is directed at specific times, with multiple opportunities to be directed to a waste stream marked 'Drain'. Up to 1 liter is collected in each bottle within the carousel. The two refrigerated parts of the system house the Accumulator and the Carousel, where the sample is stored for up to 3 days before delivery to an analytical laboratory for analysis.

called an accumulator. At regular intervals, the accumulator then dumps into 1 liter sample bottles contained in a carousel
within a refrigerator below the accumulator (Baumgardner et al., 1997).

Deployment of the collector was automated starting in 1994 during the MADPro campaign, and, since that time, the following meteorological parameters have been used to trigger deployment of the collector: 1) liquid water content (LWC) above
$0.05$ g m$^{-3}$ measured by a Gerber Particle Volume (PVM) (Gerber, 1991), indicating the presence of a cloud, 2) temperature above 2°C to prevent damage to the collector from riming, 3) presence of rain not detected, limiting measurements to
105 non-precipitating clouds. An Aerochem gridded rain sensor was used from 1994 to 2014, which was updated in 2015 to a CAPMoN heated grid rain sensor (Mekis et al., 2018), 4) and lastly, wind-speed above 2 m s$^{-1}$, as measured using a RM Young anemometer, allowing cloudy air to pass through the collector. If all of these meteorological parameters are met for at least 1 minute, the collector is raised from its protective housing, exposing the Teflon strings to the passing airflow. If the state

meteorological parameters are not met for at least 1 minute, the collector returns to its protective housing. A rain valve in line with the accumulator is instantaneously activated (within 1 second) if rain is detected, sending rain water to waste. Figure 1 summarizes the major events and changes to the cloud water collection at WFM.

The automated collection system originally deployed the cloud water collector using an electric garage door opener motor, which was upgraded in 2005 to a pneumatic system to increase deployment and retraction speed. A pressurized tank of deionized water also replaced the manual electric pump spray system in 2005. To monitor for potential contamination, blanks and rinses of the cloud collector were obtained regularly throughout each summer and assessed for the same analytes as the cloud water samples. The collector was rinsed with deionized water for at least 15 seconds between samples, except when sample changes occurred while deployed in cloud.

Cloud water samples were refrigerated at temperatures of 4°C for up to 3 days within the carousel refrigerator and then transported to a laboratory for subsequent chemical analysis. From 1994-2006, the accumulator dumped into a sample bottle every hour and the carousel advanced to the next bottle. Starting in 2007, samples were dumped every hour and later pooled together into 3 hour composite samples prior to analysis. In 2014, the one hour samples were pooled together into 12 hour composite samples. Then, starting in 2015, the 1 liter accumulator was replaced with a 12 liter accumulator to accumulate cloud water over 12 hour periods, providing up to one daytime and one nighttime sample per day. The new 12 liter accumulator was housed in a separate refrigerator above the carousel refrigerator with the intent of preventing microbial degradation prior to freezing the sample, and a measurement of the accumulator weight was added to continuously monitor the volume of cloud water collected.

Control of the cloud water collection system is accomplished through four pinch valves, three of which divert the sample stream one of two ways under different conditions, and a pneumatic 'Dump valve' to start or stop flow out of the accumulator. The 'Collector valve' diverts flow to waste whenever the collector is not actively deployed, primarily for the purpose of diverting cleansing spray water to the waste stream. The 'Drain valve' diverts flow to waste after 1 liter has been collected in sample bottles within the carousel (since bottles in the carousel can only hold 1 liter). The 'Rain valve' operates at a higher frequency than the 1 minute timescale of the 'Collector valve' and diverts flow to waste when the rain sensor detects rain but the collector has not yet retracted back into its housing.

Starting in 2018, an automated filtering system with in-line peristaltic pump was added just prior to the accumulator (Fig. 2). This change in protocol was introduced in an attempt to further safeguard samples for subsequent analysis of the organic constituents, since organics now comprise a major fraction of aerosol and cloud water solute mass, and their chemical and physical properties remain poorly understood. Automated filtering could be important for preserving some organic compounds for up to 3 days within the carousel refrigerator, since some microbes are known to both consume and produce organic compounds even at temperatures as low as 0 °C (Van Stempvoort and Biggar, 2008). Note that for TOC, microbial degradation is only important if it leads to organic compounds volatilizing out of solution. For the first year of automated filtering, only a few samples were filtered during collection, and field tests were conducted to verify that the filter used did not introduce measurement artifacts. It should be noted that after automated filtering commenced, TOC could no longer be measured, as filtration removes insoluble organic carbon, and measurements of water soluble organic carbon (WSOC) were obtained, instead. To evaluate the potential

impact of this change in protocol, both TOC and WSOC were measured for a subset of samples in 2018 and 2019, and these re-
sults indicated on average 15% insoluble organic carbon (Fig. S2), consistent with previous studies at other locations (Herckes
et al., 2013).

As used by Khwaja et al. (1995), a pre-washed $0.4\mu$m track-etched polycarbonate filter was selected, to limit potential
degradation of organics but with low enough pressure drop to attain reliable operation of the automated filtering apparatus.
The in-line filter was housed within the accumulator refrigerator to further limit microbial growth on the filter substrate. Since
pressures in excess of 40 psig could occur under high particulate load on the filter, PTFE reinforced silicone peristaltic pump
tubing (GORE STA-PURE PCS) was used to prevent tubing rupture, and filters were replaced during every site visit, nominally
every 3 days. The peristaltic pump was also triggered to only turn on during cloud events when the collector was deployed, to
prevent unnecessary wear and tear on the peristaltic pump tubing.

## 2.2 Chemical Analysis and Data Handling

The list of routinely measured analytes for this long-term dataset is: conductivity, pH, $SO_4^{2-}$, $NO_3^-$, $NH_4^+$, chloride ($Cl^-$),
sodium (Na), calcium (Ca), magnesium (Mg), and potassium (K), with TOC measurements beginning in 2009 and WSOC
measurements beginning for a subset of samples in 2018 and the majority of samples 2019-2021. From 1994-2007, nitrite
($NO_2^-$) was also measured. However, concentrations were rarely above detection limits and, on average, $NO_2^-$ contributed less
than 1 $\mu$eq $L^{-1}$ to the ion balance. Due to these reasons, $NO_2^-$ was no longer included in the suite of measured analytes. The
measurement techniques, method detection limits, precision and accuracy for the MADPro, ALSC and AWI measurements are
summarized in Table S1 and further described by Isil et al. (2000) and Lee et al. (2022). Measurements conducted by AWI
in 2018 and 2019 are described here. Sulfate and $Cl^-$ concentrations were measured via ion chromatography using a Lachat
QC 8500 Ion Chromatograph. A cadmium reduction technique coupled with a Lachat QC 8500 Flow Injection Analyzer was
used to measure $NO_3^-$, as described by the EPA 352.2 method. It should be noted that cadmium reduction measures both $NO_2^-$
and $NO_3^-$, potentially biasing $NO_3^-$ high. AWI measured $NO_2-$ concentration for a subset of 14 cloud water samples in 2018
using a Dionex Integrion HPIC System and found that on average $NO_2-$ was still only 2% of $NO_3-$ concentrations. For
$NH_4^+$ measurements, samples were injected with an alkaline solution of sodium hydroxide, converting all $NH_4^+$ to $NH_3$. The
$NH_3$ was then separated from the solution using a hydrophobic semi-permeable membrane, which was then combined with a
solution containing a pH indicator and measured continuously at 590nm using a flow photometer. WSOC was measured using
a Shimadzu TOC-VCPN Total Organic Carbon Analyzer. Accuracy was determined using prepared reference standards and
were run every 10 cloud samples. Metals including Ca, Mg, Na, and K were measured using Inductively Coupled Plasma -
Atomic Emissions Spectrometry (ICAP-AES).

It is important to note that the methods used by MADPro, ALSC, and AWI measure total metal concentrations rather than
their ionic forms, which could potentially overestimate the role of metals in the overall charge balance. Samples for the majority
of 2019 and 2020 and all of 2021 were filtered during the collection process, removing potential insoluble Ca and Mg larger
than $0.4\mu$m, and yet reductions in the concentrations of Ca and Mg were not observed, in spite of the relatively high pH
often encountered in recent years. This suggests that a large fraction of Ca and Mg in the historical cloud water dataset was

also dissolved. The assumption has long been that all measured Ca and Mg were fully dissolved within cloud water due to the dilute aqueous solutions that comprise cloud droplets. To evaluate whether this assumption has merit, 10 unfiltered cloud water samples collected 2018-2020 were reanalyzed by the Lance Lab using Metrohm 761 Compact Ion Chromatograph with a Metrosep C Supp 2 - 150/4 cation column. The samples were re-analyzed twice; once unfiltered and once filtered using a 0.4 $\mu$m polycarbonate filter. Figure S4 compares the concentrations of $Ca^{2+}$ and $Mg^{2+}$ before and after filtering while Fig. S5 compares the filtered $Ca^{2+}$ and $Mg^{2+}$ measurements to the original unfiltered elemental Ca and Mg concentrations measured by AWI or ALSC. For both sets of figures, the slopes of the regression lines ranged from 0.971 to 1.08 with linear correlation coefficient 0.94-0.98, indicating that filtration removed little to no insoluble Ca or Mg. To further evaluate what fraction of Ca or Mg we expect to be dissolved in the cloud water, the solubility of $CaCO_3$ and $MgCO_3$ were calculated as a function of pH (Fig. S6). These calculations show that for the full range of pH and dilution experienced by our cloud water samples to date, $CaCO_3$ and $MgCO_3$ are expected to be completely dissolved. Only at pH values of 8 and greater do we expect these species to remain in the solid phase. Other forms of calcium and magnesium, e.g. $CaCl_2$, $Ca(NO_3)_2$ or $Ca(OH)_2$, are even more soluble and even more likely to be completely dissolved at the pH and liquid water contents encountered at WFM. Based on the IC analysis and calculations described above, we therefore assume that the measured metal concentrations are equivalent to their ionic concentrations, as commonly assumed in past analyses of this dataset.

Another common practice, made by both MADPro and ALSC, was to categorize samples based on ion balance criteria according to the relative percent difference (RPD) between the measured cation and anion concentrations, calculated as follows:

$$\text{RPD} = 100\% * \frac{\sum \text{Cations} - \sum \text{Anions}}{(\sum \text{Cations} + \sum \text{Anions})/2} \tag{1}$$

Samples were categorized as Valid based on RPD using the following thresholds: RPD < 100% when both $\sum$Cations and $\sum$Anions < 100 $\mu$eq L$^{-1}$, or RPD < 25% when either $\sum$Cations or $\sum$Anions > 100 $\mu$eq L$^{-1}$. Samples that did not match these criteria were considered Invalid. Samples with flagged data due to suspected measurement problems were also considered Invalid, which were extremely rare, only representing < 1% of samples. Samples for which all analytes could not be measured (for instance, due to insufficient sample volume) were omitted from our separate analysis of Invalid and Valid datasets, and we refer to these samples as Unclassified since ion balance could not be fully assessed for these samples. Fewer than 10% of samples were Unclassified. Previous analyses involving the MADPro and ALSC data sets typically only used Valid data (i.e. those samples that passed the ion balance criteria and were also not subject to any known or suspected measurement problems), which has important implications for the data interpretation and the resulting long-term trends, as will be discussed in the results section of this work.

Previous studies also often focused on reporting sample volume-weighted analyte concentrations due to a focus on wet deposition, whereas we report measured cloud water concentrations and cloud water loadings, the latter of which complements measurements of aerosol loadings. Since a typical cloud droplet is more likely to evaporate than deposit to the surface at any given time (Seinfeld and Pandis, 2016), this latter focus on atmospheric loadings is useful for investigating chemical processes occurring in the atmosphere separately from the influence of sample dilution at higher LWC.

To account for variability in LWC that potentially affected cloud water concentrations, cloud water loadings (CWL) are calculated as follows (Elbert et al., 2000; Marinoni et al., 2004; Kim et al., 2019):

$$\mathrm{CWL} = \frac{\mathrm{LWC}_{\mathrm{samp}} * [\mathrm{X}_i] * \mathrm{mw}_i}{\rho_{\mathrm{w}} * \mathrm{Z}_i} \tag{2}$$

where CWL is expressed in $\mu$g solute per m$^3$ of air, $\mathrm{LWC}_{\mathrm{samp}}$ is the representative LWC during which the sample was collected expressed in g water per m$^3$ of air, $[\mathrm{X}_i]$ is the liquid concentration of a given solute ion $i$ expressed in $\mu$eq solute per L of water, $\mathrm{mw}_i$ is the molecular weight of the solute in g mol$^{-1}$, $\mathrm{Z}_i$ is the solute charge and $\rho_w$ is the density of cloud water (assumed to be 1 g cm$^{-3}$). To calculate TOC CWL, $[\mathrm{X}_i]/\mathrm{Z}_i$ is replaced with molar concentration in $\mu$mol C per L and $\mathrm{mw}_i$ = 12g mol$^{-1}$. While the long-term WFM data set included average LWC values for each sample, there is no record of how these values were determined, and careful analysis indicates that these values included LWC values during periods with drizzle when collected cloud water was being sent to a waste stream and the collector is not deployed, a common occurrence during the night. To avoid introducing bias into the CWL calculation, we recalculated $\mathrm{LWC}_{\mathrm{samp}}$ by removing LWC values from the average when the rain sensor detected rain for > 15% of a given measurement period or the collector is deployed for the less < 25% of a measurement period as described in the supplemental material (Fig. S7). Since this additional information about the collector status and meteorological conditions during each hour is only available starting in 2009, only data from 2009 to 2021 is included in our $\mathrm{LWC}_{\mathrm{samp}}$ and CWL calculations.

In the present study, data analysis is performed within the statistical software R (Team, 2021). Measured analyte concentrations and conductivity exhibit a log-normal distributions, therefore median values were used rather than means for trend analysis, as the median is more robust to outliers and a better representation of the population. Additionally, due to significant variance within the trend data, Mann-Kendall (MK) trend tests combined with Theil-Sen slope estimators (Sen, 1968) were employed to obtain a robust estimate of the slope of the long-term trends in measured analyte concentrations, with its significance determined by the calculated p-value from MK test.

## 3   Long-Term Trends

### 3.1   Trends in Valid Data

This section updates the long-term trends in cloud water composition at WFM that were previously reported by Aleksic et al. (2009), Dukett et al. (2011) and Schwab et al. (2016b), now including data through 2021. Figure 3a shows annual median analyte concentrations, and the slopes of the linear trend analysis for the Valid dataset, while Table S2a shows the slopes, and associated p-values for all measured analytes. Annual median cloud water pH increased from 3.78 in 1994 to 5.34 in 2021, corresponding with a substantial decrease in conductivity and $SO_4^{-2}$ concentrations. A similar trend was found in data reported by Pye et al. (2020) for cloud water pH from the 1990s to the 2010s while showing data as far back as the 1970s being highly acidified and remaining flat until the 1990s. Meanwhile, $NO_3^-$, $NH_4^+$, and $Cl^-$ concentrations exhibited relatively modest decreases, which leveled off starting in 2006. The remaining ions exhibited no discernible trends.

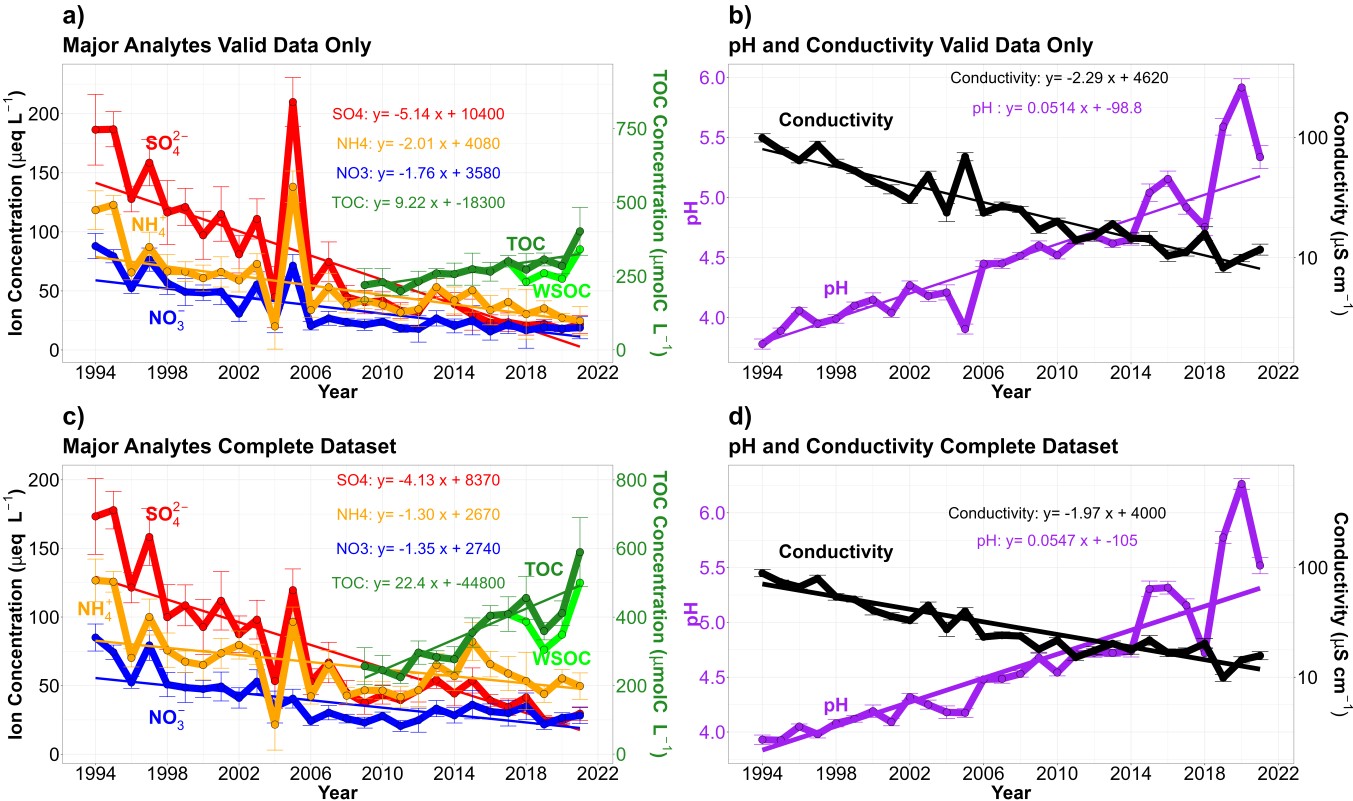

**Figure 3.** Annual median trends in cloud water collected from the summit of WFM. Plots a and b include Valid data, samples achieving an approximate ion balance, while plots c and d include the entire available dataset. Annual median trends in major analyte concentrations reported in $\mu$eq L$^{-1}$ for ions on the left axis, and $\mu$molC L$^{-1}$ for TOC on the right axis. Conductivity is measured in $\mu$S cm$^{-1}$ on the right axis in plots b and d. Measured annual median WSOC for samples collected in 2018-2021 are shown in light green for comparison to the estimated TOC concentrations in these years. Error bars represent the standard error of the annual mean concentrations. Standard error of the annual mean were used instead of interquartile range as the large sample to sample spread in the cloud water composition produced large error bars, making the figures difficult to read.

TOC is the only analyte that shows evidence of an increasing trend (plotted on the right axis in Fig. 3a). This is the first reported long-term trend for cloud water TOC at WFM. It should be noted that for 2009, the first year of routine TOC measurements, TOC analysis was only conducted on 20 samples in total, compared to 80-285 samples per year with TOC measurements from 2010-2021. In spite of the small samples size for 2009, these data are included in the trend analysis, as there is no evidence of measurement error or bias, and their inclusion doesn't strongly impact the resulting trend. Since TOC concentrations could not be measured for samples when automated filtering was performed during collection (as applied for samples 2018-2021), annual median TOC concentrations for these 4 years were estimated based on the average 85% WSOC/TOC ratio observed for the subset of samples from 2018 and 2019 where both WSOC and TOC were measured (Fig. S2). Both WSOC and TOC

concentrations are shown in Fig. 3a for these four years, but only the estimated TOC concentrations are used in the trend analysis.

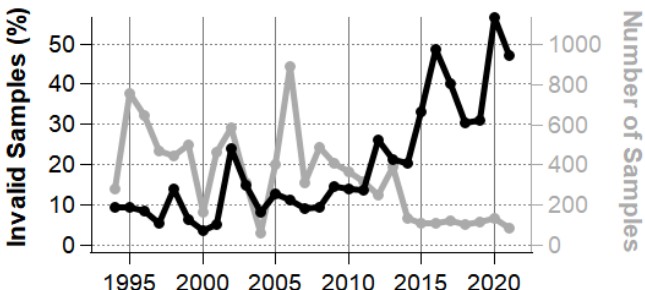

**Figure 4.** Annual percentage of cloud water samples considered Invalid according to ion balance criteria [Invalid/(Invalid + Valid)] (black, left axis) and total number of samples (Valid + Invalid + Unclassified) in each year (grey, right axis).

## 3.2 The Growing Influence of Invalid Data

Review of the cloud water measurements obtained at WFM shows that the proportion of so-called Invalid samples has increased substantially over the past several years, increasing from 9.3% in 1994 to 47% in 2021, and peaking at 57% in 2020 (Fig. 4).
While there is a seasonality to the percentage of Invalid samples, every summer month shows an increasing slope (Fig. S3). This classification, based on ion balance criteria, was originally intended to reduce the impact from measurement error, as it is assumed that positively charged cations should be balanced by negatively charged anions in any natural bulk water systems. Implicit in this characterization is the assumption that the majority of ions are being analyzed for. This growing ion imbalance in cloud water coincides with increasing cation/anion ratio observed in rain water collected by the National Atmospheric
Deposition Program (NADP) over the same time period throughout NY State (Rattigan et al., 2017) and more broadly across the eastern U.S. and Canada (Feng et al., 2021), based upon the same standard suite of inorganic analytes. Since the rain water observations are entirely independent of the cloud water measurements, it is highly unlikely that the growing number of Invalid samples is due to a measurement artifact. Crucially, aside from ion imbalance, there is no evidence that the so-called Invalid cloud water samples were subject to measurement bias and should be discarded. We hypothesize, instead, that the growing ion
imbalance is due to unmeasured ions like organic acids that are growing in abundance, as also hypothesized by Dukett et al. (2011) and Feng et al. (2021). Since organic acids are known to comprise a significant fraction of TOC (Herckes et al., 2013), the measured trend in TOC concentrations at WFM supports this hypothesis.

Since the Invalid data set now comprises nearly half of the samples it is worth analyzing this dataset separately. These samples are potentially representing a chemically distinct subset of samples that are encountered more frequently in recent
270 years. To summarize, Invalid samples are considerably less acidic (0.5-1 pH units) and exhibit higher concentrations of $NH_4^+$, TOC, $Ca^{2+}$ and $K^+$ (Fig. S8), and the trends are significantly different as well. After around 2006, trends in the Invalid dataset

appear to begin diverging from the Valid data set. On average, LWC is higher within the Valid data set, which tends to dilute samples and thus reduce all analyte concentrations. However, there is not a growing divergence between the trends of Valid and Invalid LWC, therefore LWC trends alone do not explain the divergence between Valid and Invalid concentrations. These observations show that while the Invalid samples have been growing more common, the composition of the Invalid samples has also grown more concentrated in TOC, $NH_4^+$, $NO_3^-$ and $Ca^{2+}$ over the past decade.

## 3.3 Trends of the Complete Data Set

Including Valid, Invalid and unclassified data, the trends still exhibit significant reductions in conductivity, $SO_4^{2-}$, $NO_3^-$ and $NH_4^+$ concentrations (Fig. 3b and Table S2b). However the reductions are lower in magnitude compared to only the Valid data. For instance, the slope of the regression line for $SO_4^{2-}$ concentrations decreases from 5.14 to 4.13 $\mu$eq L$^{-1}$ yr$^{-1}$. Notably, the reactive nitrogen species ($NH_4^+$ and $NO_3^-$) exhibit a more noticeable uptick in concentration from 2007 to 2014 before decreasing again. This apparent inflection point in the trends, starting around 2006 appears to signify the emergence of a new chemical regime that has been growing in importance along with the growing fraction of Invalid samples. The biggest impact of including the Invalid data in our analysis is a stronger increasing trend in TOC concentrations, which more than doubles from 2009 to 2021. TOC shows a strong seasonality, typically with June and July exhibiting the highest concentrations for the year (Fig. S9). Theil-sen regression analysis reveals a statistically significant increasing trend for TOC in the months of June, July, and September. No other analyte measured in WFM cloud water exhibit any significant seasonality during these four summer months. Annual median concentrations for the complete dataset show that $NH_4^+$ has been increasingly capable of neutralizing $SO_4^{2-}$ and $NO_3^-$, which is well matched by a decreasing trend in measured H$^+$ (Fig. 5). Note that for individual cloud water samples, $NH_4^+$ concentrations can be greater than both $SO_4^{2-}$ and $NO_3^-$ concentrations combined (in terms of $\mu$eq L$^{-1}$), which occurred for 1203 out of 9429 cloud water samples from 1994 to 2021.

The increasing trend in annual median Cation/Anion ratios within cloud water over the entire record is 1.97% yr$^{-1}$ (p = < 0.001), including only samples where cations and anions for all measured analytes are reported. Emergence of a new chemical regime within the past decade is further supported by these observations, with annual median ion imbalance < 20% prior to 2009, after which median ion imbalance has exceeded 20% in every subsequent year except 2018.

## 4 Relationship Between Cloud Chemistry and Liquid Water Content

The analysis so far has focused on cloud water concentrations, which can be impacted by changes in meteorological parameters like LWC that are unrelated to changes in chemistry or emissions. Previous works have investigated the role of LWC on analyte concentrations (Elbert et al., 2000; Aleksic and Dukett, 2010; van Pinxteren et al., 2016), finding a weak non-linear negative relationship between total ion concentrations (TIC) and LWC, with considerable variance within the data. Aleksic and Dukett (2010) used a probabilistic technique to describe the relationship between binned LWC and TIC, with TIC showing a negative exponential relationship with LWC. When using this type of LWC binning technique for TOC, a similar probabilistic

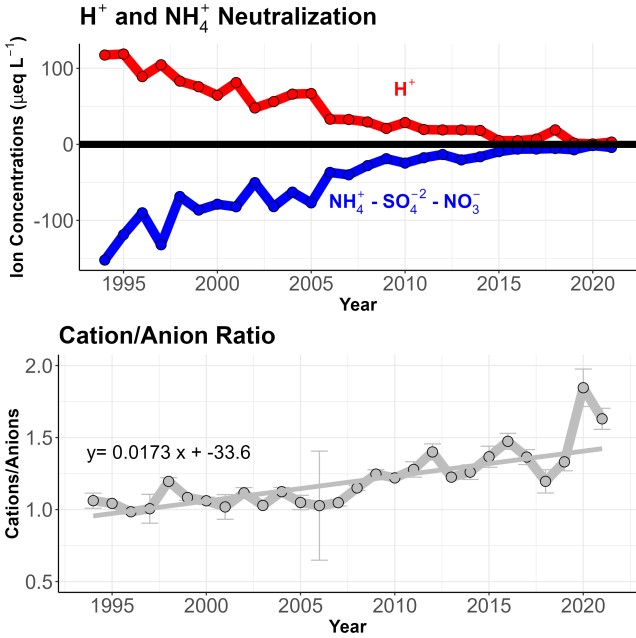

**Figure 5.** a) Annual median measured $H^+$ concentrations and neutralization of $SO_4^{2-}$ and $NO_3^-$ by $NH_4^+$, and b) annual median Cation/Anion ratios, in cloud water collected from the summit of WFM (complete dataset).

relationship can be found (Fig. S10), where a decreasing trend is observed in TOC as LWC increases, but exhibiting large variability within each bin, similar to what has been found previously at other locations (Herckes et al., 2013).

An alternate way to report cloud and precipitation composition is to weight by sample volume (Schwab et al., 2016a; Rattigan et al., 2017), which provides a means for determining the net accumulation of a particular analyte in the environment. Sample volume for cloud water is dependent on the collection efficiency of the cloud water collector, which is influenced by wind speed, LWC and updraft, complicating the relationship between sample volume and cloud composition. In our analysis, instead of sample volume, samples are weighted by liquid water content (LWC) to determine Cloud Water Loadings (CWL),

also referred to as air-equivalent concentrations (Marinoni et al., 2004; Wang et al., 2016; Kim et al., 2019), which has the added advantage of being directly comparable to aerosol loadings.

Figure 6 and Table S3 show the trends and the associated p-values in median CWL for 2009-2021, a time period for which meteorological data was available for calculating the average in-cloud LWC for each cloud water sample within the complete dataset. Focusing on this date range has the added advantage of focusing on a time period that is chemically distinct from the

earlier half of the long-term record (e.g. with $NH_4^+$ concentrations growing in abundance relative to both $SO_4^{2-}$ and $NO_3^-$, and changing trends in all the major analytes, as shown in Fig. 3c). Using the same analysis method as in previous sections, increasing trends in $Ca^{2+}$, $Mg^{2+}$, $Na^+$, $Cl^-$, and $K^+$ CWL since 2009 are observed. The trends in $Ca^{2+}$ and $Mg^{2+}$ CWL imply increased influence from mineral or soil dust, while the increasing trends in $Na^+$ and $Cl^-$ suggest increasing influence from sea spray or playa dust. The increasing $K^+$ trend could indicate increased influence from wildfires (Simoneit, 2002;

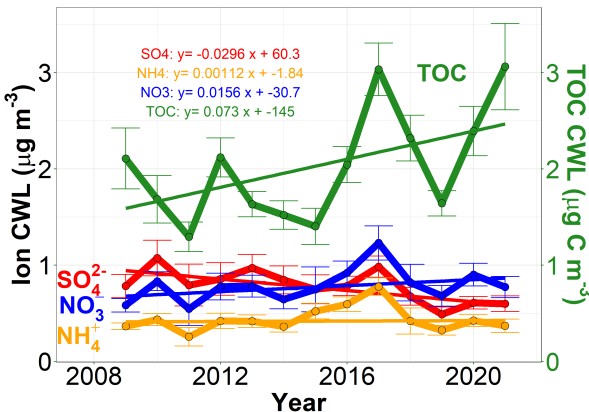

**Figure 6.** Linear trends in annual median Cloud Water Loadings (CWL) for the major inorganic ions and TOC between 2009-2021 for the complete data set, using the meteorology criteria outlined in the Supplement, Section S5 to calculate average in-cloud LWC values associated with each sample.

Pachon et al., 2013), though $K^+$ can also be found in mineral dust. Meanwhile, there is no statistically significant trend in LWC, $SO_4^{2-}$, $NH_4^+$, $NO_3^-$ or TOC CWL over this time period. This surprising result indicates that the large increasing trend in TOC concentrations since 2009 reported in Fig. 3c is not statistically significant when accounting for LWC, despite the positive slope of the TOC CWL regression line. Additional measurement uncertainty introduced by the LWC measurements may partly account for the higher p-value and reduced significance of the TOC CWL trend, but complex relationships between

LWC and TOC concentrations (e.g. dilution and partitioning of soluble organic gases and/or drier air masses associated with greater biomass burning emissions) could also be playing a role.

## 5    Potential Missing Ions

Two independent measures (Cation/Anion ratio and Predicted/Measured Conductivity) imply that as pH increases, there is a growing abundance of analytes that are not being measured, particularly above 4.5 pH (Fig. S11). The measured ion imbalance

suggests that more of the missing analytes are anions than cations. Here, we discuss potential missing anions responsible for the increasing trend in cation/anion ratios in WFM cloud water.

### 5.1    Bicarbonate

Due to increasing cloud water pH, bicarbonate ($HCO_3^-$) is hypothesized to have a growing contribution to ion balance. The dissolution of $CO_2$ is a major source of $HCO_3^-$ by partitioning into cloud droplets and hydrolyzing to form carbonic acid

($H_2CO_3$). This contribution of $HCO_3^-$ from $CO_2$ increases as pH increases until the pH approaches the pKa of $CO_3^{2-}$, where the influence of $HCO_3^-$ then decreases. Highly alkaline dust particles containing calcium and magnesium carbonate ($CaCO_3$ and $MgCO_3$) can also be a major source of $HCO_3^-$. Cloud water collection sites that are heavily influenced by mineral dust

often report pH values above 7 (Khemani et al., 1987; Budhavant et al., 2014), leading to relatively acid-buffered cloud water. While the suite of measurements at WFM did not include $HCO_3^-$, it can be estimated using pH dependent equilibrium equations for an aqueous system open to the atmosphere, calculated as follows:

$$[HCO_3^-] = \frac{P_{CO2} * H_{CO2} * Ka_1}{[H^+]} \tag{3}$$

where $P_{CO2}$ represent the partial pressure of $CO_2$, assumed to be 410 ppm, $H_{CO2} = 3.2x10^{-2}$ M atm$^{-1}$ is the Henry's law constant of $CO_2$ and $Ka_1 = 4.37x10^{-7}$ (Sander, 2015). Temperature was also assumed to be 298 K. There is virtually no change to the overall ion balance when including estimated $HCO_3^-$ (Fig. S12). Even for pH values above 6 (approximately the $pK_a$ of $H_2CO_3$), there are only modest improvements in overall ion balance. Based on these estimates, it is unlikely that $HCO_3^-$ is a major driver of ion imbalance in the WFM cloud water observations. Importantly, these are only estimates of $HCO_3^-$ concentrations based on equilibrium with the atmosphere. There has been significant discussion on the validity of Henry's Law constants for cloud drop distributions, as several field studies note significant departures of measurements from gas-aqueous phase partitioning (Pandis and Seinfeld, 1991; Winiwarter et al., 1994; Straub, 2017). It is plausible that some deviations could also exist for the carbonate system, thus leading to inaccurate $HCO_3^-$ concentration estimates. One fog water monitoring site in Pennsylvania found similar anion deficiency in their system and hypothesized $HCO_3^-$ as one potentially missing anion. For better estimates, they used inorganic carbon measurements from a TOC analyzer and assumed all inorganic carbon was part of the carbonate system. They then used the pH of the fog water to calculate the fraction of $HCO_3^-$. This led to around a 2.4x greater concentrations of $HCO_3^-$ on average than predictions based on Henry's Law, improving overall ion balance (Straub, 2017). It should be noted that the fog water exhibited considerably greater pH values than WFM cloud water, however, and only fog water samples with pH > 6.0 showed appreciable ion imbalance, reinforcing our assessment that $HCO_3^-$ concentrations are likely negligible for the majority of cloud water samples from WFM.

## 5.2 Organic Anions

Organic anions such as formate, acetate, and oxalate are known to be present in cloud and rain water, including at WFM (Khwaja et al., 1995; Chapman et al., 2008; Herckes et al., 2013; Akpo et al., 2015). A strong correlation between TOC and ion imbalance ($R^2 = 0.55$, $p < 0.001$) implies that an important fraction of missing anions are likely organic (Fig. S13). Further evidence for missing organic acids is revealed when investigating ion balance as a function of pH. At low pH, the ratio of measured Cations to Anions is approximately 1, approaching ion balance. As pH increases, the ratio decreases, reaching a minimum of 0.69 for pH 5.5-6.3, with large decreases in the ratio at pH bins of 4.0-4.8 and 4.8-5.5 (Fig. S14). These pH bins are close to the pKa values of formic and acetic acid (3.74 and 4.75 respectively), often the most abundant organic acids found in cloud water (Herckes et al., 2013). As pH increases, a greater fraction of these acids, and others, are expected to exist in their anionic form, and contribute more significantly to charge balance. Increasing cloud water pH also increases the fraction of volatile and semi-volatile organic acids that reside in the aqueous phase, contributing further to the ion imbalance (Seinfeld and

Pandis, 2016). Similarly, Fig. S11 shows that the ratio of predicted/measured conductivity decreases as pH increase, indicating
a missing source of ions within the system that could be organic in nature.

Assuming that the missing anions are organic acids, the contribution of organic acids to TOC can be approximated. We introduce the value of a carbon to charge ratio (C/z) to represent how many carboxylic acid functional groups are associated with every carbon atom in an organic molecule. Figure S15 shows the annual median faction of organic acids to TOC for 4 different C/z ratios of 1, 2, 3, and 5. C/z = 1 represents the smallest mono and dicarboxylic acids formic and oxalic acid. C/z =
2 represents slightly larger but commonly found compounds like acetic and succinic acid. C/z = 3 represents compounds like propionic and lactic acid. Lastly, C/z = 5 is used to represent large organic compounds like humic-like substance (HULIS), a class of compounds that are not well defined but often represent a large fraction of WSOC (Perminova et al., 2003; Graber and Rudich, 2006; Spranger et al., 2020). A representative HULIS compound with a molecular weight of 360 g mol$^{-1}$, an organic matter/organic carbon ratio of 2, and 3 carboxylic acid functional groups indicates a molecule with 15 carbon atoms
and a a C/z value of 5. These values are estimated based on available literature (Perminova et al., 2003; Graber and Rudich, 2006; Salma and Láng, 2008; Brege et al., 2018; Cook et al., 2017; Qin et al., 2022). Ionic organic compounds represent an important fraction of TOC, which we estimate range from 4.81-24.0% in 2009 to 7.28%-36.4% in 2021, and potentially peaking at 58.0% in 2020. There is also strong evidence that this fraction is increasing as indicated by theil-sen regression (p = 0.0327), indicating that not only is TOC increasing but organic ions are also increasing. There is very little data to constrain
high molecular weight organic compounds in WFM cloud water. One study byCook et al. (2017) measured high molecular weight organic compounds for eight cloud water samples collected at WFM in 2014 using electrospray ionization coupled with Fourier transform ion cyclotron resonance mass spectrometry (FTICR-MS). This work found that the distribution of the carbon numbers were largely centered around 15 atoms, which is in rough agreement with our estimate of 3 carboxylic acid groups and a molecular weight of 360 g mol$^{-1}$. Therefore, the C/z ratios of 1 and 5 may serve as useful lower and upper estimates
for the fraction of TOC that is ionic. More detailed chemical analysis is required to better estimate the contribution of organic anions to the chemical system.

## 6  A New Chemical Regime

### 6.1  Changing Relationship Between H$^+$ and Conductivity

While our analysis of the long-term cloud water data set at WFM illuminates important differences from previous analyses,
there is no doubt that significant progress has been made in reducing ambient concentrations of criteria pollutants across the U.S. and at WFM since the 1990s, resulting in significant decreases in $SO_4^{2-}$ and $NO_3^-$ concentrations, with associated increases in cloud water pH and decreases in conductivity. These overall trends highlight that, generally, conductivity is controlled by H$^+$ concentrations, due to the high limiting molar conductivity of H$^+$ ($\Lambda^+$ = 349.5 S cm$^2$ mol$^{-1}$) compared to other analytes (e.g. 160.0 S cm$^2$ mol$^{-1}$ for $SO_4^{2-}$ and 73.5 S cm$^2$ mol$^{-1}$ for $NH_4^+$) (Coury, 1999). In recent years, this relationship is diminishing.
Figure 8a shows the relationship between conductivity and pH for individual cloud water samples throughout the entire 28 year record (1994-2021). pH values range from <3 to nearly as high as 7, and conductivity values likewise span nearly 3 orders

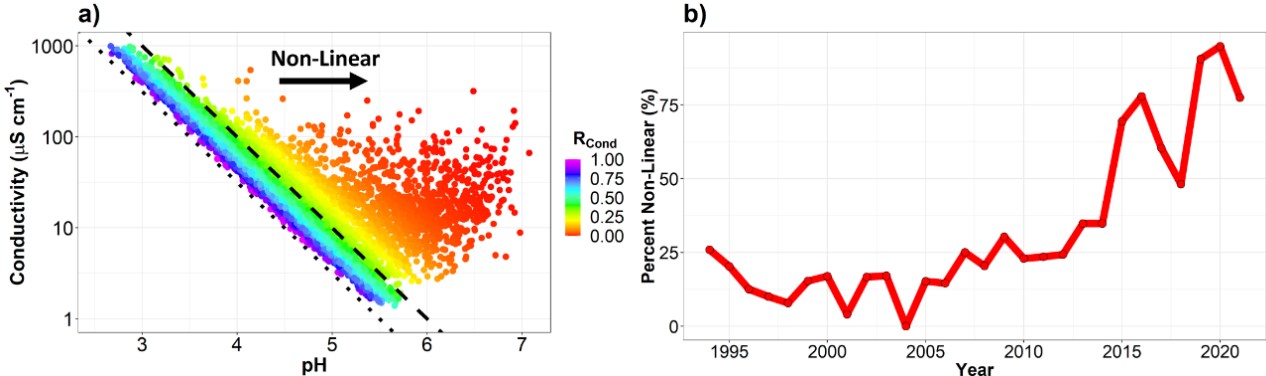

**Figure 7.** a) Conductivity and pH for individual cloud water samples from 1994 to 2021. The Linear and Non-Linear regimes are distinguished with dashed line that indicates $H^+$ contribution to the measured conductivity is exactly 35%. To the right of this line, $H^+$ contributes less to the measured conductivity, and we refer to this as the Non-Linear Regime, since conductivity and pH no longer exhibit a clear linear relationship. The dotted line represents the conductivity if $H^+$ is the only ion present. b) Percent of samples in the Non-Linear Regime for each year of sampling at WFM.

of magnitude. Many samples exhibit a strong linear relationship between $H^+$ and conductivity, especially in more acidified cloud samples. However, beginning at pH values above 4, many samples deviate from this linear relationship, with higher conductivity values than expected for samples with high pH. To reach these same conductivity values, the cloud water samples

requires much higher concentrations of other ions such as $NH_4^+$ and $Ca^{2+}$. The emergence of this new 'Non-Linear Regime' with both high pH and high conductivity implies that one cannot assume that samples with higher pH are 'cleaner', as was typically the case in decades past at WFM. To better investigate this non-linear relationship between conductivity and $H^+$, we propose a new method of classifying cloud water data as follows:

$$R_{Cond} = \frac{H_{Cond}}{M_{Cond}} \tag{4}$$

where $M_{Cond}$ is the measured specific conductivity and $H_{Cond}$ is the conductivity from the $H^+$ ion, calculated from:

$$H_{Cond} = \Lambda^+ [H^+] \tag{5}$$

where $\Lambda^+$ is the limiting molar conductivity of the $H^+$ ion at infinite dilution arising from the auto-ionization of pure water. If $R_{Cond} < 0.35$, less than 35% of the total conductivity of the sample can be explained by $H^+$. For samples with $R_{Cond} > 0.35$, to the left of the dashed line on Fig. 7a, a linear relationship between conductivity and pH is observed, and for samples with

$R_{Cond} < 0.35$ this linear relationship breaks down. The following section will compare differences in chemistry between the two regimes.

**Dunn-Test Comparison of Regimes**

| Species | Non-Linear Median | Linear Median | Difference | Percent Difference | p-value | Non-Linear n | Linear n |
|---|---|---|---|---|---|---|---|
| TOC (umolC/L) | 424.641 | 236.067 | 188.574 | 44.408% | p < 0.001 | 745 | 836 |
| NH4 (ueq/L) | 77.767 | 55.513 | 22.255 | 28.616% | p < 0.001 | 1873 | 6357 |
| SO4 (ueq/L) | 42.890 | 86.653 | -43.763 | -102.035% | p < 0.001 | 1863 | 6377 |
| Ion Balance (ueq/L) | 37.514 | 6.999 | 30.515 | 81.343% | p < 0.001 | 1823 | 6172 |
| NO3 (ueq/L) | 33.310 | 36.382 | -3.072 | -9.222% | 0.823 | 1874 | 6377 |
| Ca (ueq/L) | 28.202 | 5.725 | 22.477 | 79.700% | p < 0.001 | 1866 | 6350 |
| Conductivity (uS/cm) | 21.210 | 40.600 | -19.390 | -91.419% | p < 0.001 | 1879 | 6431 |
| Mg (ueq/L) | 6.144 | 1.811 | 4.333 | 70.524% | p < 0.001 | 1865 | 6329 |
| pH | 5.285 | 4.190 | 1.095 | 20.719% | p < 0.001 | 1879 | 6431 |
| Cl (ueq/L) | 2.962 | 2.426 | 0.536 | 18.096% | p < 0.001 | 1862 | 6354 |
| Na (ueq/L) | 1.740 | 0.968 | 0.772 | 44.368% | p < 0.001 | 1852 | 6342 |
| K (ueq/L) | 1.540 | 0.928 | 0.612 | 39.740% | p < 0.001 | 1853 | 6291 |
| LWC (g/m^3) | 0.425 | 0.570 | -0.145 | -34.118% | p < 0.001 | 1811 | 5976 |

**Table 1.** Linear versus Non-Linear Regime median concentrations for all measured analytes and associated p-values for the Dunn test.

## 6.2  Linear Regime vs Non-Linear Regime

There are many similarities between the Valid/Invalid and Linear/Non-Linear classifications. Figure 7b shows that a growing fraction of samples are in the Non-Linear Regime, peaking at over 90% of samples in 2020. Table 1 uses a Dunn test to compare median concentrations between the Linear and Non-Linear regimes (averaged over all years for which data is available, i.e. 1994-2021 for all analytes except for TOC, which was averaged over 2009-2021). Conductivity and $SO_4^{2-}$ are nearly double and cloud water pH is 1 unit lower in the Linear data set compared to the Non-Linear data set, highlighting the dominance of $SO_4^{2-}$ and $H^+$ in the Linear Regime. There is no obvious difference in $NO_3^-$ concentrations between the two data sets, implying that the controlling factors for $NO_3^-$ are unrelated to the relationship between $H^+$ and conductivity. Every other measured ion shows higher concentrations in the Non-Linear data set than the Linear data set, with considerably higher concentrations of TOC, $Ca^{2+}$, and $Mg^{2+}$, and in addition, an overall greater ion imbalance. Therefore, the Linear Regime can be characterized as a highly acidified system controlled by $SO_4^{2-}$, and the Non-Linear Regime is a system with greater TOC, $NH_4^+$ and base cations. The Non-Linear Regime is becoming the predominate system in WFM cloud water.

## 6.3  Is Cloud Water Representative of Cloud Droplets?

With the reductions in $SO_4^{2-}$ and $NO_3^-$, species such as $Ca^{2+}$ have grown in relative importance to ion balance, conductivity, and pH. Sources of atmospheric $Ca^{2+}$ can include fossil fuel combustion in the form of fly ash (Lee and Pacyna, 1999), but the dominant source is generally thought to be mineral dust particles containing bases like calcium carbonate ($CaCO_3$) and calcium oxide (CaO) (Reff et al., 2009). These particles typically raise the pH of cloud and rain water, particularly evident

in observations from India (Khemani et al., 1987; Budhavant et al., 2014). Within WFM cloud water, there is a considerably higher concentration of $Ca^{2+}$ for samples within the Non-Linear Regime, contributing to the higher measured pH values.

Due to the mechanism by which they are lofted into the atmosphere, mineral dust aerosols are known to exist primarily in the coarse mode (i.e., with dry particle diameter $> 1\mu m$) (Seinfeld and Pandis, 2016). While these aerosol can make up a substantial fraction of the overall aerosol mass concentration, they generally represent a very small fraction of the aerosol number concentration (Mahowald et al., 2014). Therefore, even if these particles are hygroscopic, only a small fraction of activated cloud droplets are likely affected by coarse mode aerosol, even at a remote location like WFM where aerosol concentrations are relatively low. During a pilot study at WFM in 2017, cloud droplets intercepted at the summit of WFM were shown to be primarily comprised of cloud condensation nuclei with 100-300nm dry diameter (Lance et al., 2020), consistent with this general understanding.

When the rare alkaline cloud droplets, formed on coarse mineral dust aerosol, are collected by the cloud water collection system, they can have an out-sized impact on the bulk cloud water sample, which may no longer well represent the majority of cloud droplets as they existed in the cloud. This is supported by work from Moore et al. (2004) using a multistage cloud collector at WFM, which found that $Ca^{2+}$ and $Mg^{2+}$ concentrations were 2-30 and 2-58 times greater, respectively, in large droplets than small droplets. Size resolved aerosol composition data in nearby Ontario, Canada (VandenBoer et al., 2011) indicates that 65-95% of the total aerosol $Ca^{2+}$ mass and 78-99% of $Mg^{2+}$ were within super micron ($> 1 \mu m$) aerosol in 2009-2010 when those measurements were conducted, while > 99% of the total number of aerosol were found in sub micron particles (Fig. S16).

Based on this general understanding of coarse mode aerosol, as discussed above, we expect that bulk collection of cloud water is often no longer representative of the vast majority of cloud droplets as they existed in the atmosphere and could bias our understanding of cloud droplet pH. In an attempt to better account for this potential bias, a new calculation for inferred pH of a typical cloud droplet ($pH_{TD}$) is introduced:

$$pH_{TD} = -\log_{10}([H^+] + [Ca^{2+}] + [Mg^{2+}]) \tag{6}$$

where $[H^+]$, $[Ca^{2+}]$ and $[Mg^{2+}]$ are the measured concentrations in units of eq $L^{-1}$ within bulk cloud water. We refer to this as the "top down" (TD) approach for estimating acidity of the majority of cloud droplets because it is based on the measured bulk cloud water pH, which is assumed to provide the cumulative impact from all dissolved ions (identified or not). By accounting for measured $Ca^{2+}$ and $Mg^{2+}$ concentrations, this calculation is an attempt to remove the influence of mineral dust particles on the bulk cloud water pH, assuming that measured Ca and Mg concentrations are associated with dissolved alkaline compounds such as $CaCO_3$ or $MgCO_3$ in equilibrium with atmospheric $CO_2$ and that a very small number of droplets contain these minerals at all. These assumptions are supported by the assessment of Ca and Mg soluble fractions reported in Section S4 of the Supplement and the size-resolved aerosol composition measurements described in Section S13 of the Supplement. Additional measurements (e.g. size resolved aerosol and/or cloud droplet residual composition collocated with the cloud water measurements) would be needed to better constrain the cloud droplet pH estimate. While the measured pH

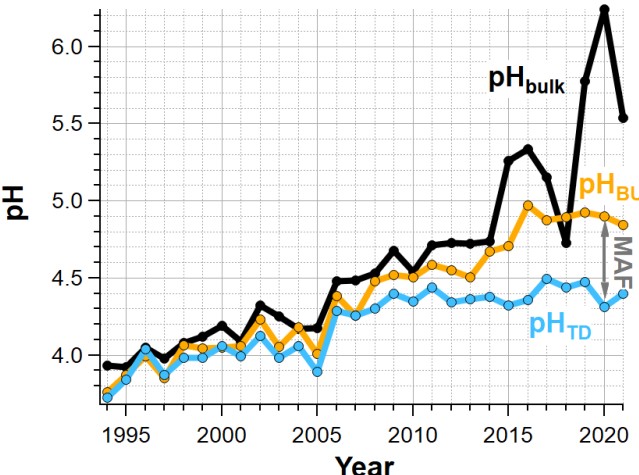

**Figure 8.** Annual median bulk cloud water pH (black) compared to estimated cloud droplet pH calculated from the bulk cloud water pH and measured $Ca^{2+}$ and $Mg^{2+}$ concentrations ($pH_{TD}$, blue) and estimated cloud droplet pH calculated from measured $SO_4^{2-}$, $NO_3^-$ and $NH_4^+$ concentrations ($pH_{BU}$, orange).

of bulk cloud water is relevant for wet deposition, we propose that $pH_{TD}$ is more relevant for processes occurring within the majority of cloud droplets as they existed in the atmosphere.

Figure 8 shows the annual median estimated cloud droplet pH ($pH_{TD}$, blue) and measured bulk cloud water pH (black). During the first ~10 years of the cloud water monitoring program, $pH_{TD}$ increases in parallel with $pH_{bulk}$, but the influence of base cations on measured pH has grown over time. Despite measured pH continuing to increase, there has been little change in $pH_{TD}$ since ~2010. In 2020, $Ca^{2+}$ and $Mg^{2+}$ are calculated to have increased the median cloud water pH from 4.3 to 6.2, resulting in nearly two orders of magnitude decrease in $H^+$ concentrations.

To further evaluate the causative factors behind the growing discrepancy between the measured bulk cloud water pH and estimated cloud droplet pH, we also calculate one of the simplest pH proxies based on measured $SO_4^{2-}$, $NO_3^-$ and $NH_4^+$ concentrations (Pye et al., 2020):

$$pH_{BU} = -\log_{10}([SO_4^{2-}] + [NO_3^-] - [NH_4^+]) \tag{7}$$

This "bottom up" (BU) approach includes the major inorganic species known to form through secondary processes, which are typically found in fine aerosol, and are therefore typically found in the vast majority of cloud condensation nuclei. Gas phase $NH_3$ and/or $HNO_3$ that dissolved into cloud droplets are also included with this proxy method. Entirely neglected in this approach are contributions from organic acids within fine aerosols, or partitioning of organic acid gases to the aqueous phase after cloud droplet activation. Also, as noted previously, a substantial fraction of cloud water samples exhibit greater $[NH_4^+]$ than $[SO_4^{2-}] + [NO_3^-]$, which corresponds to negative values for $[H^+]_{BU}$. This is a significant limitation for this proxy method in recent years, when over 40% of cloud water samples exhibit this condition (Fig. 9). Since negative $H^+$ values cannot be

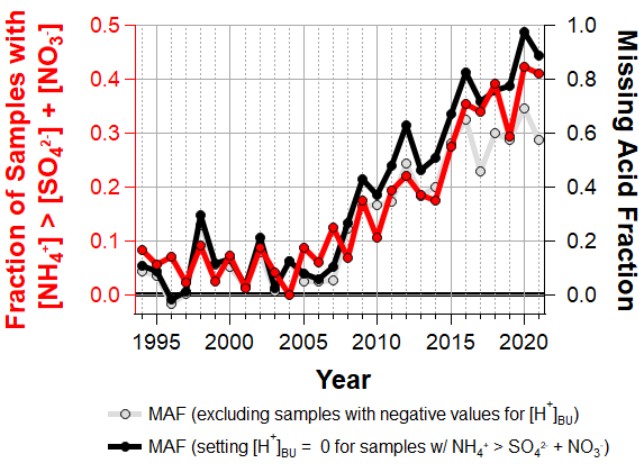

**Figure 9.** Fraction of cloud water samples where $[NH_4^+] > [SO_4^{2-}]+[NO_3^-]$ (left axis), and annual median Missing Acid Fraction (right axis).

included in the $pH_{BU}$ calculation, only the samples traditionally expected to be acidic (i.e. those with relatively high $SO_4^{2-}$ and/or $NO_3^-$) are included in the annual average $pH_{BU}$ in Fig. 8. If cloud droplets are more acidic than $pH_{BU}$, this implies that other ions within cloud droplets (other than $SO_4^{2-}$ or $NO_3^-$) are making significant contributions to the cloud droplet acidity.

Early in the cloud water monitoring program, annual median $pH_{BU}$ tracked $pH_{TD}$ well, indicating that $SO_4^{2-}$, $NO_3^-$ and $NH_4^+$ concentrations alone were able to capture the estimated cloud droplet pH (within 0.2pH units through 2009). In recent years, a growing discrepancy between annual median $pH_{BU}$ and $pH_{TD}$ is observed, with as much as 0.61 pH difference observed in 2016. These results indicate that $SO_4^{2-}$, $NO_3^-$ and $NH_4^+$ are increasingly unable to explain the estimated cloud droplet pH.

In calculating $pH_{TD}$, it is important to note that even though pH can shift dramatically when accounting for the influence from rare alkaline droplets, the cation/anion ratio remains exactly the same as it was for bulk cloud water since the increased $H^+$ is exactly balanced by the removal of the two base cation species ($Ca^{2+}$ and $Mg^{2+}$). Therefore, the reassessment of bulk cloud water data in terms of inferred cloud droplet acidity does not resolve the ion imbalance problem.

The top down and bottom up approaches described above provide two independent estimates for cloud droplet pH. By comparing these two proxies, we then infer the fraction of cloud droplet acidity that is unexplained by the major inorganic ions. We define the Missing Acid Fraction (MAF) as:

$$MAF = \frac{[H^+]_{TD} - [H^+]_{BU}}{[H^+]_{TD}} \qquad (8)$$

When MAF is zero, the measured $SO_4^{2-}$, $NO_3^-$ and $NH_4^+$ concentrations are sufficient to explain 100% of the inferred cloud droplet acidity, and when MAF is 1.0, the measured $SO_4^{2-}$, $NO_3^-$ and $NH_4^+$ concentrations explain 0% of the inferred cloud droplet acidity. Fig. 9 shows the annual median MAF with two different ways of handling samples with negative values for $[H^+]_{BU}$. First, samples with negative values for $[H^+]_{BU}$ were excluded from the calculation (plotted in grey). Out of concern that a majority of cloud water samples now exhibit $[NH_4^+] > [SO_4^{2-}]+[NO_3^-]$, another calculation for MAF was performed

by setting $[H^+]_{BU}$ to zero for this subset of samples, as $SO_4^{2-}$ and $NO_3^-$ are completely neutralized by $NH_4^+$ under these conditions (plotted in black). Prior to $\sim$2008, $[SO_4^{2-}]+[NO_3^-]-[NH_4^+]$ was able to capture the inferred cloud droplet acidity typically within about 20%. Since then, a rapid increase in the contribution of unmeasured ions to cloud droplet acidity is inferred from this dataset, with more than half of the inferred cloud droplet acidity left unexplained in the last several years.

### 6.4 Implications for Cloud Droplet pH

The large and growing discrepancy between estimated cloud droplet pH and measured $SO_4^{2-}$, $NO_3^-$ and $NH_4^+$ concentrations over the past decade in cloud water at WFM coincides with a growing number of samples with higher conductivity than can be explained with the measured inorganic analytes (Fig. S11) and a growing abundance of cloud water organic carbon. Altogether, these independent measures provide strong support for the hypothesis that organic acids are growing in importance and now have a significant impact on cloud droplet acidity (further discussed in Section S14 of the Supplement). This is a 515 major departure from the original intent of the collection site to monitor and investigate inorganic acid deposition.

In addition to increasing TOC concentrations, there is evidence that a growing fraction of TOC is ionic since the ion imbalance is growing faster than the increase in TOC concentrations (Fig. S15). Given that the proportion of organic acids in their deprotonated state is dependent on pH, and yet the estimated cloud droplet pH has been relatively flat for the past 15 years, the increasing ion imbalance suggests that organic acids could be increasing in total concentration as well as relative importance. 520 Finally, annual median cloud droplet pH averages 4.4 for the years 2009 to 2021, which is close to the pKa for formic acid and acetic acid (pKa = 3.74 and 4.75, respectively). These two organic acids have been found to be the most abundant organic acids in cloud and rain water (Khwaja, 1995; Keene and Galloway, 1986; Paulot et al., 2011; Herckes et al., 2013; Millet et al., 2015; Pye et al., 2020), but will contribute much less to ion balance at a pH of 4.4 than the annual median bulk cloud water pH of >5.5 in recent years would suggest. Measured concentrations for a larger range of organic acids than typically analyzed for 525 may be necessary to capture the important chemical species controlling cloud droplet pH.

A number of assumptions must be recognized when interpreting results from this simple analysis. For a more precise estimate of typical cloud droplet acidity, other ions associated with coarse mode aerosol, like $NO_3^-$, would also be included in the $pH_{TD}$ calculation. However, given that $NO_3^-$ may be found predominantly in coarse or fine aerosol at different times (VandenBoer et al., 2011), the contribution from coarse mode $NO_3^-$ is not sufficiently constrained by the measurements avail- 530 able. To the degree that $NO_3^-$ is present in coarse mode aerosol along with $Ca^{2+}$ and $Mg^{2+}$, the calculation for $pH_{TD}$ would be an underestimate for cloud droplet pH. Similarly, $pH_{BU}$ would be underestimated to the same degree, since a fraction of $NO_3^-$ would not have been present in cloud droplets that formed on fine aerosol, as had been assumed. A combination of $NO_x$ reductions and potentially increasing mineral dust aerosol within the region may be changing the fraction of $NO_3$ within fine aerosol, which would need to be better constrained to improve our $pH_{BU}$ and MAF estimates. 535 Also implicitly assumed is that LWC associated with coarse mode aerosol is negligible for both the $pH_{TD}$ and $pH_{BU}$ proxies of cloud droplet pH. We argue this is a decent assumption since, while the coarse mode aerosol can contribute significant aerosol mass in spite of extremely low number concentrations due to their much larger size (1-2 orders of magnitude larger diameter for the coarse mode than the accumulation mode), the relative contribution of liquid water from cloud droplets formed

on coarse mode aerosol is much lower due to the inverse relationship between condensational growth rate and droplet diameter (i.e. droplets originating from the coarse and accumulation modes are likely to have diameters within a factor of two of each other, for non-precipitating clouds (Seinfeld and Pandis, 2016)).

Conversely, we argue that using the bulk cloud water pH as a constraint for simulations of gas/droplet partitioning and aqueous chemistry, without accounting for the aerosol mixing state, can lead to substantial biases for the vast majority of cloud droplets as they existed in the atmosphere. Size-resolved measurements of $Ca^{2+}$, $Mg^{2+}$ and $NO_3^-$ in aerosol and/or cloud droplet residuals are required to confirm the validity of the $pH_{TD}$ estimates reported here in order to confidently and accurately evaluate these types of atmospheric processes. But, inferring cloud droplet pH as we do in this paper may yield a much better representation of a typical cloud droplet than the bulk cloud water measurements do, in the present day.

While we cannot directly evaluate the validity of our simple calculation for $pH_{TD}$ with the current suite of measurements, it is encouraging that several important relationships are substantially simplified when evaluated against $pH_{TD}$ instead of the measured bulk cloud water pH. In particular, the relationship between $pH_{TD}$, LWC and $SO_4^{2-}$ concentrations is much simpler and easier to explain (Fig. S18). For a given LWC, there is a clear inverse relationship between $SO_4^{2-}$ and $pH_{TD}$, which maxes out at $pH_{TD}$ ~5.6 at very low $SO_4^{2-}$ concentrations, coinciding with the pH expected for a pure water droplet in equilibrium with atmospheric $CO_2$. In contrast, $SO_4^{2-}$ concentrations exhibit a much more chaotic relationship with measured bulk pH values.

The relationship between TOC concentrations and $pH_{TD}$ is also substantially simplified. Figure 10 shows that TOC concentrations are much better correlated with $pH_{TD}$ values throughout the entire data set (2009-2021), as compared to measured bulk cloud water pH. This might imply that, like $SO_4^{2-}$, organic compounds within cloud water are tightly linked with the cloud droplet pH. This finding may be consistent with the majority of organic aerosol mass being of secondary origin (Jimenez et al., 2009), since several known mechanisms for secondary organic aerosol (SOA) formation include acid dependent reactions (Hallquist et al., 2009; Tilgner et al., 2021). However, the other major secondary aerosol species including $NH_4^+$ also show simplified relationships with $pH_{TD}$ as compared to measured bulk cloud water pH (Fig. S19), which may imply that the emissions sources or formation mechanisms for these chemical species are linked in other ways, in tandem with their known prevalence in fine aerosol.

## 6.5  Driving Factors for Increasing TOC

The growing abundance of TOC in cloud water (both in relative and absolute terms) may be at least partially due to an increase in partitioning of volatile low molecular weight organic acids to the condensed phase as a result of lower cloud droplet acidity (Tilgner et al., 2021). However, the strong inverse relationship between TOC and estimated cloud droplet pH (Fig. 10a) appears to undermine this hypothesis. Furthermore, $SO_4^{2-}$ concentrations and $pH_{TD}$ trends have been relatively flat in recent years, in contrast to the rapid increase in TOC concentrations, which shows no signs of slowing down. There is also evidence that cloud water TOC in the late 1980's and early 1990's, when cloud droplet acidity was very high, were similar to concentrations today (Khwaja et al., 1995; Anastasio et al., 1994), which would also suggest a decoupling of $SO_4^{2-}$ and TOC trends.

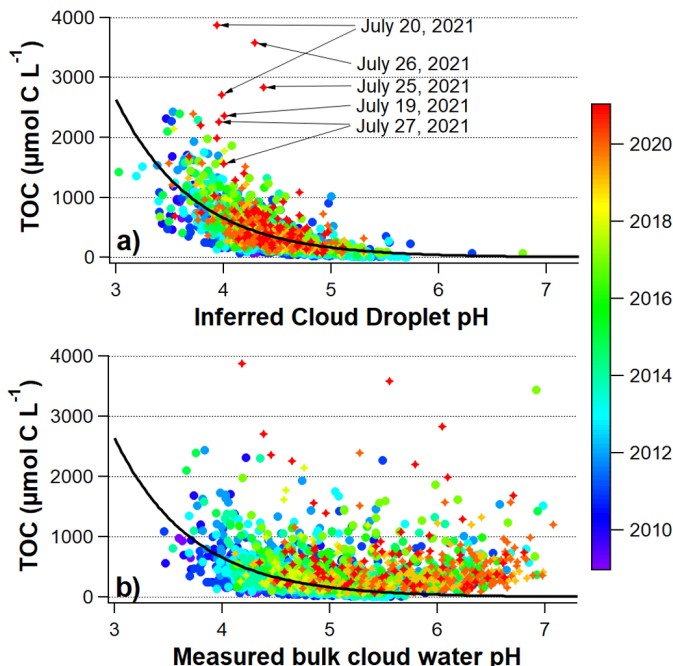

**Figure 10.** Measured TOC concentrations for individual samples 2009-2017 (circles) and measured WSOC concentrations for individual samples 2018-2021 (stars), versus a) inferred cloud droplet pH ($pH_{TD}$), and b) measured bulk cloud water pH, colored by the year that the cloud water sample was collected at WFM. The exponential curve in both a) and b) is the fit to all data in a): $TOC_{fit} = A exp(-pH/\tau)$, $A = 1.7193 \times 10^5 \pm 2.6 \times 10^4$, $1/\tau = 1.3934 \pm 0.038$

Another potential driving factor for the increasing TOC trend is an increase in wildfire smoke, which can be an important source of both primary and secondary organic matter in both aerosol and cloud water, even after multiple days of transport downwind (Cook et al., 2017; Di Lorenzo et al., 2018; Zhou et al., 2017; Lee et al., 2022). Figure 10a highlights several cloud water samples with exceptionally high measured WSOC concentrations coinciding with a haze event across the northeastern U.S. in July 2021 associated with wildfire smoke originating in south central Canada and the northwestern U.S. (Shrestha et al., 2022) (further described in Section 16 of the Supplement). Black carbon and carbon monoxide measurements from the summit of WFM were also unusually high (e.g. as high as 1780 ng m$^{-3}$, compared to a more typical value of 100 ng m$^{-3}$ for black carbon). The relatively close proximity of some of these fires to WFM may have have been partly responsible for the exceptionally high WSOC concentrations and the strong deviation from the well established correlation with $pH_{TD}$, possibly due to an unusually large contribution from freshly emitted primary organic matter. Episodic events with exceptionally high concentrations such as these are the reason we report annual median rather than annual mean concentrations for the long-term trends. The increasing median trend of K$^+$, particularly since 2009 (Fig. S8), is potential evidence for increased smoke influence, as K$^+$ is often used as a passive biomass burning tracer (Reid et al., 2005).

## 7 Summary and Conclusions

This work updates the long-term trend analysis of cloud water chemistry monitoring at Whiteface Mountain (WFM) over the past 28 years, with critical review of past methodologies resulting in significant changes to the data interpretation and additional analysis of TOC trends since 2009 when TOC was first routinely measured at WFM. In the past, many samples were excluded from analysis if they did not achieve an approximate ion balance from measured species. Using this criteria, a growing number of samples were being excluded from the data set, peaking at 57% of the samples in 2020, with no evidence of measurement error. When evaluating the complete data set, decreases in $SO_4^{2-}$, $NH_4^+$ and $NO_3^-$ become more modest, organic carbon increases at a much faster rate, and increasing trends in $Ca^{2+}$ and $Mg^{2+}$ emerge. The increasing fraction of samples that do not achieve ion balance is associated with an increasing inorganic cation/anion ratio, implying there are anions that are not being measured with the long-term measurement suite, which are growing in abundance. When evaluating changes in cloud water loading (CWL), the trend in organic carbon is no longer significant, indicating that much of the trend in TOC was associated with variability in LWC. The increasing trend in $Ca^{2+}$ CWL remains significant.

When investigating potential missing anions, $HCO_3^-$ and organic acids were identified as two important candidates. Estimates of $HCO_3^-$ using cloud water pH, and atmospheric $CO_2$ indicate that for most cloud water samples, $HCO_3^-$ is negligible. For samples with pH > 6, which are growing more common, $HCO_3^-$ could be important. However, samples with bulk cloud water pH this high are often likely impacted by externally mixed alkaline droplets formed on coarse mineral dust aerosol that don't represent the majority of cloud droplets as they existed in the atmosphere. Organic acids are therefore likely more important than $HCO_3^-$ for explaining ion imbalance for the majority of cloud droplets originating from fine (submicron) cloud condensation nuclei. VandenBoer et al. (2011) found that organic acids were found predominantly within submicron aerosol, which supports this hypothesis. The strong correlation between organic carbon and inorganic ion imbalance also supports this hypothesis, since organic acids are known to comprise a significant fraction of organic carbon.

In the early years of the cloud water monitoring program, conductivity was largely controlled by $H^+$ due to its high concentrations and large molar conductivity relative to other ions in the system. In recent years, conductivity is no longer linearly proportional to $H^+$ concentrations. We use the break down of the linear relationship between conductivity and $H^+$ to characterize a new chemical regime that is less acidic but exhibits relatively high conductivity. A growing fraction of samples are now being classified in this Non-Linear Regime, peaking at over 90% of samples in 2020. The Non-Linear Regime is characterized by higher concentrations of TOC, $NH_4^+$, and base cations, particularly $Ca^{2+}$.

Altogether, the reduction of $SO_4^{2-}$ and $NO_3^-$, and increases in $Ca^{2+}$ and $Mg^{2+}$ have resulted in a situation where bulk cloud water pH likely does not represent the majority of cloud droplets as they existed in the atmosphere, since $Ca^{2+}$ and $Mg^{2+}$ are believed to primarily reside in coarse aerosol that only represent a small fraction of cloud droplets. To account for this bias, we proposed a new calculation for "Estimated cloud droplet pH" that accounts for the measured Ca and Mg concentrations. While measured bulk cloud water pH has steadily increased for the entire duration of routine cloud water monitoring program, estimated cloud droplet pH has remained flat since 2009, implying that much of the increasing cloud water pH in recent years is driven by $Ca^{2+}$ and $Mg^{2+}$ rather than the reductions in $SO_4^{2-}$.

A growing number of samples exhibit concentration of $NH_4^+$ that are greater than the sum of $SO_4^{2-}$ and $NO_3^-$ concentrations, providing further evidence for a major missing source of acidity. For the samples where it is possible to predict cloud droplet acidity from the measured $SO_4^{2-}$, $NO_3^-$ and $NH_4^+$ concentrations, a growing discrepancy is found in comparison to the estimated cloud droplet pH, with the majority of the inferred cloud droplet acidity unexplained by the measured $SO_4^{2-}$, $NO_3^-$ and $NH_4^+$ concentrations in the past several years.

Considerable changes have occurred in cloud water at WFM over the past 28 years of monitoring, from a system dominated by $SO_4^{2-}$ to a system controlled by base cations and organic compounds. Many other regions in the world are seeing changes as $SO_2$ emissions continue to decrease. This type of system is considerably less studied, with important implications for air quality, ecosystem health, and climate. More research, in combination with additional measurements of aerosol and trace gases, are required to better understand the system. As WFM and potentially other regions of the world enter this new chemical regime, additional measurements will be needed to study the effect on important processes like secondary organic aerosol production and nitrogen deposition.

*Competing interests.* All authors declare no competing interests.

*Code and data availability.* Cloud water data can be found at http://atmoschem.asrc.cestm.albany.edu/~cloudwater/pub/Data.htm. The code and data used to make the figures can be found at https://doi.org/10.5281/zenodo.7379622

*Author contributions.* J. Dukett and P. Snyder conducted the cloud water sampling and chemical analysis 2001-2017. P. Casson, R. Brandt, C. Lawrence and S. Lance conducted the cloud water sampling 2018-2021. D. Kelting and L. Yerger conducted the chemical analysis 2018-2019. P. Snyder conducted the chemical analysis 2020-2021. C. Lawrence and S. Lance analyzed the data sets. C. Lawrence carried out the statistical analyses. C. Lawrence and S. Lance wrote the manuscript with contributions from all co-authors.

*Acknowledgements.* Cloud water measurements reported in this paper were supported by the New York State Energy Research and Development Authority (NYSERDA) under the Adirondack Long Term Monitoring project (2001-2017) and NYSERDA Contract 124461 (2018-2021). Previous cloud water measurements (1994-2000) conducted by MADPro were funded by the U.S. EPA under Contract 68-02-4451 and 68-D2-0134. WFM trace gas and meteorological measurements were supported by NYSERDA Contract 48971. NYSERDA has not reviewed the information contained herein, and the opinions expressed in this report do not necessarily reflect those of NYSERDA or the State of New York. P. Casson and S. Lance thank Eric Hebert for helping to set up the cloud water collection system. S. Lance and C. Lawrence thank Gabriele Pfister and Rajesh Kumar of NCAR for providing output from the WRF-Chem forecast.

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
