# Peer review of "Long-Term Monitoring of Cloud Water Chemistry at Whiteface Mountain: The Emergence of a New Chemical Regime"

_Atmospheric Chemistry and Physics, 2022_

## Author Response (AR1)

**Responses to Reviewer 1**

We wish to express our sincere appreciation to the reviewer for their thorough review, which we believe has substantially improved the paper.

**Abstract, lines 5-7: This sentence seems largely irrelevant to the abstract.**

This sentence was removed from the abstract.

**Abstract, line 12: I suggest referring to an "inorganic" ion balance.**

Inorganic was added to the term ion balance. This is an important clarification as organic ions were not measured in the cloud water samples.

**In the abstract and throughout the manuscript the authors refer to measurements of Ca2+ and Mg2+; however, the measurement techniques they report measure elemental Ca and Mg, not the cationic form. The cation concentrations will be equal to or lesser than the elemental concentrations. This reporting issue can be easily corrected in the manuscript, but that does not solve the problem of using these species in calculating the ion balance or the inferred pHTD (eqn. 6).**

This is a legitimate concern and we thank the reviewers for bringing this to our attention. Note that for all cloud water samples that were filtered during collection (as we have been routinely doing since taking over the cloud water monitoring program in 2018), we can be more confident that the reported measurements for calcium and magnesium are cationic, since any insoluble particles >0.4um would have been removed through filtration prior to analysis.  Also note that the most recent years are the most impacted by calcium (with the greatest measured cation to anion ratio and the greatest difference between the measured bulk cloud water pH and estimated cloud droplet pH), in spite of filtration.

However, for the majority of the long-term monitoring program, samples were not filtered. The assumption has long been that all measured calcium and magnesium were fully dissolved within cloud water due to the dilute aqueous solutions that comprise cloud droplets. This assumption has been "baked in" to every previous analysis conducted with this dataset to our knowledge (including Aleksic et al., 2009; Schwab et al., 2016; Pye et al., 2020; Lee et al., 2022), given that only the "ion balanced" samples (including Ca and Mg, assumed to be in their ionic form) have been labeled as "Valid".  Since this is an important assumption behind our analysis as well, we tested this assumption by conducting additional analysis of 10 unfiltered cloud water samples that were archived from 2018-2020. Only a small number of samples could be re-analyzed in this way, because we do not have archived cloud water samples prior to 2018

when routine filtering began.  To our knowledge, this sort of test was never conducted on samples from the historical dataset. For our reanalysis of these samples, we used a Metrohm 761 Compact Ion Chromatography (IC) system to measure cationic $Ca^{2+}$ and $Mg^{2+}$. We re-analyzed these samples twice; once without filtering and once filtering the samples through a 0.4 μm polycarbonate filter. We found that there were virtually no differences between the filtered and unfiltered IC measurements.  When compared to the original unfiltered elemental measurements, we also found very little difference (over a wide range of pH values, 4.5-6.4), consistent with the measured Ca and Mg being completely dissolved for these samples.

In the supplemental material, we also show that, if $Ca^{2+}$ and $Mg^{2+}$ are associated with $CaCO_3$ and $MgCO_3$, we expect them to be completely dissolved for the full range of pH and dilution that have been observed for cloud water samples collected at WFM.  Note that other forms of calcium and magnesium, e.g. $CaCl_2$, $Ca(NO_3)_2$ or $Ca(OH)_2$, are even more soluble and even more likely to be completely dissolved at the pH and liquid water contents encountered at WFM. Detailed information about the IC analysis and the $CaCO_3$ and $MgCO_3$ solubility calculations as a function of pH described above can be found in section 4 of the supplemental material.

Aleksic, N., Roy, K., Sistla, G., Dukett, J., Houck, N., & Casson, P. (2009). Analysis of cloud and precipitation chemistry at Whiteface Mountain, NY. Atmospheric Environment, 43(17), 2709-2716. https://doi.org/10.1016/j.atmosenv.2009.02.053

Lee, J. Y., Peterson, P. K., Vear, L. R., Cook, R. D., Sullivan, A. P., Smith, E., et al. (2022). Wildfire smoke influence on cloud water chemical composition at Whiteface Mountain, New York. Journal of Geophysical Research: Atmospheres, 127, e2022JD037177. https://doi.org/10.1029/2022JD037177

Pye, H. O., Nenes, A., Alexander, B., Ault, A. P., Barth, M. C., Clegg, S. L., ... & Zuend, A. (2020). The acidity of atmospheric particles and clouds. Atmospheric chemistry and physics, 20(8), 4809-4888. https://doi.org/10.5194/acp-20-4809-2020

Schwab, J. J., Casson, P., Brandt, R., Husain, L., Dutkewicz, V., Wolfe, D., ... & Dukett, J. E. (2016). Atmospheric chemistry measurements at Whiteface Mountain, NY: Cloud water chemistry, precipitation chemistry, and particulate matter. Aerosol and Air Quality Research, 16(3), 841-854. https://doi.org/10.4209/aaqr.2015.05.0344

**The authors correctly point out the important variations that often exist among the pH of individual cloud droplets. In calculating the inferred pH, they attempt to remove one factor that changes the alters the pH of drops that form on coarse mode dust aerosol.  They neglect, however, that other factors will still contribute to variations in pH among other, remaining cloud drops.**

Coarse mode aerosol is chemically distinct from fine mode aerosol due to the very different pathways that give rise to their presence in the atmosphere.  This is one of the reasons that policy makers often distinguish between PM1 or PM2.5 and PM10. While imperfect, it is often useful to classify the aerosol types in this way.  We attempted to remove the influence of coarse mode aerosol on bulk cloud water because the coarse mode aerosol is so chemically distinct (as supported by the size-resolved aerosol composition measurements discussed in the paper) and comprises a very small fraction of the aerosol number concentration (and therefore does not represent a "typical" cloud droplet). There are other processes such as gas-aqueous partitioning or aqueous chemistry that can impact the pH of cloud droplets, but those processes are likely to act on all droplets to some degree (and the degree to which they act on the droplets depends, to first order, on whether those droplets formed from fine mode or coarse mode aerosol particles).

**Line 34: Sulfate is not necessarily formed photochemically in the aqueous phase as many of the key reactions can occur in the dark. The oxidants (e.g., H2O2, O3) may have previously formed photochemically. It would be simpler to just say SO2 is oxidized to form sulfate.**

The sentence was changed to simplify, as recommended by the reviewer. The sentence was changed to:

"Through field experiments at WFM and other locations in the Eastern U.S., researchers established that $SO_2$, originating largely from fossil fuel combustion, can dissolve in cloud droplets and undergo aqueous oxidation to form sulfate ($SO_4^{2-}$)."

**Line 34: I do not think there is compelling evidence that significant nitrate is formed via aqueous phase chemistry in many environments. This sentence should just focus on sulfate.**

The reviewer is correct in that cloud droplets are not a major production pathway for nitric acid. We removed the mention of nitrate in this sentence.

**Lines 36-38: more references are needed here.**

More references were added to this section.

**Lines 71-73: Please state clearly that this result is specific to New York.**

The specification of New York state was added.

**Line 93: The cited reference (Anastasio et al., 1994) is missing from the reference list**

The citation was added to the list of references.

**Lines 97-99: Please state the materials used for the tubes and sample bottles.**

We use amber high-density polyethylene sample bottles and Bev-A-Line transfer pump tubing outdoors (which have an ethyl vinyl acetate shell with polyethylene liner) and platinum cured silicone tubing indoors.

**Line 101: Please state how long the LWC had to exceed the threshold to activate the collector. How quickly was the collector turned off when LWC dropped?**

The sentence was changed to be more specific about the timing of the deployment of the cloud water collector, as follows:

"If all of these meteorological parameters are met for at least 1 minute, the collector is raised from its protective housing, exposing the Teflon strings to the passing airflow. Similarly, if the state meteorological parameters are not met for at least 1 minute, the collector returns to its protective housing. A rain valve in line with the accumulator is instantaneously activated (within 1 second) if rain or no cloud is detected, sending rain water to waste."

**Line 114: How was sample dilution from residual rinse water accounted for? I doubt the collector dried quickly in a humid environment.**

Unfortunately, we will not be able to provide a realistic quantitative estimate of this potential artifact for the longterm dataset, since details in the setup (which have long since been modified) could have an impact on the overall system performance. To have a 10% impact on the measured concentrations, the residual rinse water would have to contribute 10% of the total sample volume. During the long-term cloud water monitoring program at WFM, samples typically require at least 100mL to conduct the full suite of ion measurements (required for ion balance calculations). Sample volumes require at least 250mL to also conduct TOC analysis. Sample volumes were typically much higher than these minimum values. In 2001 when hourly sampling protocols were in place, 63% of samples had volumes > 250mL and 89% of samples had volumes > 100mL. In 2017 when 12-hourly sampling protocols were in place, 80% of samples had volumes > 250mL and 92% had volumes > 100mL. The average sample volume collected during a one hour period was 450mL in 2001, while the average sample volume collected during a 12-hour period was 1960mL in 2017.

From our experience with the cloud water collection system in its current configuration, accidentally collecting 10mL or more of rinse water does not seem likely to happen, as supported by spray tests with cloud water analogs that we conducted in 2018 (which are described further below).

Spray tests with prepared concentrations of inorganic and organic salts were conducted by first creating an aqueous test solution, spraying the cloud water collection system (on the roof of the silo), and allowing the spray water to funnel down the tubing into sample bottles within the refrigerator as if they were cloud water samples.  These tests were performed to assess contamination and other potential artifacts from the cloud water collection system itself (including degradation from any microbes that might have been contained within the system and artifacts that might have been introduced by filtering).  We found that the most practical device for spraying the samples was a sprayer typically used for manual application of pesticides.  Other spraying devices were tested, including an airbrush, in an attempt to mimic a typical cloud droplet size distribution with droplet diameters centered around ~10-20 $\mu$m, but the finer droplets produced tended to get caught in the wind and lost.  The higher rate of collection for the larger drops produced with this sprayer meant that the water did not sit as long on the Teflon strings and other surfaces of the cloud water collection system as cloud water typically does.  Spray tests were only conducted on clear (non-cloudy) days, when winds were relatively low.  An attempt was made to uniformly coat the cloud water collector strings with the sprayed droplets.  The sprayer was first rinsed with hot water, rinsed twice with Type I water and then filled with Type I water and allowed to sit overnight (mimicking the bottle washing procedure of ALSC), before adding the test solutions.  Test solutions were prepared by weighing dry salt samples (purchased from Fischer Scientific), dissolving the salts in a known volume of Type I water and then diluting multiple times (first using a micropipette) to achieve the desired concentrations.

The goal for these tests was to control as many variables as possible to determine the impact of various aspects of sample handling on the measured concentrations of inorganic and organic compounds, in terms of both contamination (or dilution) and degradation.  Most of these tests focused on the degradation of organic acids, and we intend to report on those tests and more in a future paper. However, a set of tests highlighted below is relevant to the question of potential unintended dilution by rinse water residue in the cloud water system. An organic/inorganic solution mixture was prepared for these spray tests comprised of glucose and sodium acetate, formate, malonate, citrate, oxalate, pyruvate and $SO_4^{2-}$, with the intention to mimic a polluted cloud condition with realistic but high $SO_4^{2-}$ and TOC concentrations.  The sample volume of the aged sample was 56mL, and the other sprayed sample was of about the same volume (though the sample volume was not recorded, unfortunately), yielding a total collected volume for this spray test of about 100mL. Measurements of both TOC and $SO_4^{2-}$ remained quite constant from the prepared solution to the sprayed samples (97-98% of the SO4 was recovered and 96-100% of the TOC was recovered), indicating that artifacts like

dilution, contamination or degradation were not major factors for these analytes under these sampling conditions.

Summary of the relevant 2018 spray tests

| Date | Sample solution | Measured Conc. | Sample handling |
|---|---|---|---|
| June 26 | Org-inorg mixture | 8.27 mg/L $SO_4^{2-}$, 16.86 mg C/L | As prepared |
| June 26 | Org-inorg mixture | 8.01 mg/L $SO_4^{2-}$, 16.94 mg C/L | Sprayed, filtered thru 0.4 μm polycarbonate filter + polypropylene pre-filter |
| June 26-29 | Org-inorg mixture | 8.15 mg/L $SO_4^{2-}$, 16.18 mg C/L | Sprayed, not filtered, aged for 3 days in the accumulator fridge |

**Line 121: Is there evidence that the refrigeration "prevented" microbial degradation? This is a strong claim and contradicts your statement in line 135 that degradation occurs down to 0 C. Perhaps the refrigeration "reduced" microbial degradation.  Lines 130-131: What evidence do you have that filtration prevented microbial degradation?**

Our intention was not to claim that refrigeration or filtration completely prevented microbial degradation, but rather that those sample handling measures were put in place for that purpose.  The spray tests described previously do give some evidence that refrigeration, even in the absence of filtering, may have prevented microbial degradation of the cloud water mimic we concocted (which included organic acids and glucose, expected to be consumed by many different microbes that might have been present within the cloud water collection system). Measured TOC concentrations in the sprayed sample that sat within the carousel refrigerator for 3 days (alongside cloud water samples) remained within 0.4% of the concentration of the prepared solution and within 4% of the initially collected spray water (which was not aged). This is a very encouraging result. However, given that we are not monitoring the micro-ecology at all, this "spot check" cannot fully answer this question for the present day, and certainly not for the historical dataset when the micro-ecology may have been quite different. We modified the text to try and make this clearer.

Filtration was specifically introduced for the organic acid measurements that we recently began conducting, for which microbial degradation was a concern even with refrigeration down to 0C. Microorganisms are typically larger than 1 μm in size, and we expect the vast majority of microorganisms to be removed with the inline filtering.  As a precautionary measure, we began routinely filtering samples with a 0.4μm filter in 2018 when we began conducting routine organic acid measurements. While we do have some evidence that filtration may not be necessary to prevent microbial degradation of organic acids, we are not yet ready to report on those results, and we do not believe it would be appropriate to do so in this paper, because organic acid measurements were not conducted for nearly the entire duration of the long-term measurement program that is the subject of this paper. We have chosen not to report our

recent organic acid measurements in this paper (as explained in more detail in response to another question from this reviewer), but we do mention the automated filtering we initiated in 2018 because we wanted to describe the complete history of the cloud water collection system, including the last several years of the long-term measurement period (2018-2021). The biggest impact of the routine filtering that we are aware of, as pointed out in the paper, is the fraction of insoluble organic carbon that we are no longer able to measure.

**Section 2.2: Please add descriptions of pH and conductivity measurements. These are key to your analysis.**

Details of the pH and conductivity measurements throughout the cloud water monitoring program were added to the supplement (Section S1).

**Lines 161-162: Do you have any evidence to support the claim that nitrite is negligible compared to nitrate? Is this true now even at higher pH values where HONO solubility would increase?**

$NO_2^-$ was a regular analyte that was measured from 1994-2007. However, $NO_2^-$ was rarely above detection limits and even if $NO_2^-$ was detected, the concentrations were below 1 ueq $L^{-1}$ and therefore not contributing significantly to ion balance. Due to the difficulties in measurement and negligible contribution to ion balance, $NO_2^-$ was no longer included in the regular analytes. Even in 2018 with significantly higher cloud water pH, $NO_2^-$ is rarely visible on ion chromatographs, indicating it remains a minor species in terms of contributions to ion balance. We now discuss this in the manuscript.

**Line 156: Please define "AWI"**

A definition for AWI was added to line 79 and the Figure 1 caption.

**Section 2.2. As mentioned earlier, the atomic emission and absorption techniques utilized measure elements, not ions. This applies to Ca, Mg, Na, and K.**

Based on the IC analysis mentioned above and in Section 4 of the revised supplement, we assume that all Ca and Mg are in their cationic form.

**Section 2.2. Please add description of how NH4+ and metals were measured in the most recent years' samples collected by the Lance group.**

A section discussing the analytical techniques used by AWI was added reading:

"AWI injects the cloud sample with an alkaline solution of sodium hydroxide, converting all $NH_4^+$ to NH3. The NH3 is separated from the solution using a hydrophobic semipermeable membrane, which is combined with a solution containing a pH indicator. The solution is then measured continuously at 590nm using a flow photometer"

**Section 2.2. This section briefly described many different measurement techniques used by different investigators/labs during different time periods. As mentioned above, it is incomplete. It is also confusing and difficult to follow. I suggest adding a composition measurements table that would outline measurement methods, associated time periods, and relevant QC info (detection limits, accuracy, precision).**

We appreciate this suggestion, and now include a table in the revised supplement that summarizes the various techniques used at different stages of the cloud water monitoring program and the associated method detection limits, accuracy, and precision.

**Figure 3: The colors for TOC and WSOC are difficult to distinguish in panels a and c.**

Labels have been added to distinguish all measured analytes in Figures 3 and Figure 6

**It is probably worth mentioning that Pye et al (2020) report pH in cloudwater from WFM back to the mid-70s and that the values between the mid-70s and mid-90s are relatively flat (and acidic).**

A sentence was added that reads:

"A similar trend was found in data reported in Pye et al 2020 for cloud pH from the 1990s to the 2010s while showing data as far back as the 1970s with the cloud water being highly acidified and the trend remained flat until the 1990s."

**Please more clearly explain what the error bars represent. Mean standard error of what? And does it make sense to center such error bars on plots of median values?**

The standard error was clarified in Figure 3. Standard error is used as measure of variance rather than standard deviation. Due to large standard deviations, the error bars were quite large making the figure difficult to read. Similarly, using a log scale on the y-axis made it difficult to read the year-to-year changes in median concentrations. Therefore, we chose to use

standard error of the annual concentrations to provide a measurement of variance while retaining the readers ability to see the year-to-year changes in annual concentrations.

**Line 261: NH4+ concentrations increase then decrease in recent years. Why is just the increase mentioned?**

Thank you for bringing this to our attention. The increase mentioned here was left over from an older draft of the paper when we stopped our analysis at 2017. The recent decrease in $NH_4^+$ is now mentioned.

**Line 313: Bicarbonate increases with increasing pH to a point, then decreases as carbonate ion becomes more prevalent.**

The authors disagree with this statement. This situation would be true within a system that is not open to the atmosphere. However, as shown in chapter 4 of Snoeyink and Jenkins (1980), within an aqueous system in equilibrium with the atmosphere, increasing pH leads to increasing concentrations of both $HCO_3^-$ and $CO_3^{2-}$, even while the fraction of $HCO_3^-$ within the system decreases as pH approaches the pKa of $CO_3^{2-}$.

Snoeyink, V. and Jenkins, D.: Water Chemistry, John Wiley and Sons, Inc., 1980.

**Lines 342-343: Increasing pH does increase the abundance of the deprotonated acid anion. It also increases the effective solubility of the gas phase acids, which also raises concentrations in solution. The authors need to pay more attention to the important of the gas phase source of organic acids here and throughout the manuscript, especially for low molecular weight and highly volatile formic and acetic acids.**

A sentence was added to address the increased partitioning of weak acids into cloud water, potentially increasing ion imbalance further.

**Given the primary claim of this manuscript that organic acid anions are increasingly present in WFM cloud water, I am really puzzled that these compounds were not measured in samples collected by the authors. These are not especially difficult measurements. Having measured values would greatly strengthen the manuscript and enable a more complete assessment of ion charge balance and effect of the weak organic acids on droplet pH. Even in standard IC analyses, there may be evidence of these compounds in the anion chromatograms. Have the authors reviewed these?**

The original intention of the site was to monitor acid deposition in the eastern United States, so little attention was paid to the organic acid concentrations beyond 3 select cloud water episodes in 1987 (Khwaja et al., 1995), before the start of the routine long-term measurement program that is the subject of this paper. While we have begun measuring organic acids in recent years, these measurements are not available for the majority of the long-term dataset. As such, we believe that adding these measurements would be beyond the scope of the current paper. While organic acid measurements have been conducted by many groups for many years, we feel that substantial caution is required when conducting these measurements in a routine fashion as we have begun to do at WFM, particularly relating to the volatility and chemical reactivity of organic acids. Adequately handling all of these issues requires dedication of substantial time and discussion, which we are pursuing in a separate follow-up paper that focuses on the impact from organic acids in samples collected in the past few years. The questions arising from this long term trend analysis are the impetus for adding organic acid measurements to the routine suite of measurements.

Khwaja, H. A., Brudnoy, S., and Husain, L.: Chemical characterization of three summer cloud episodes at whiteface mountain, Chemosphere, 31, 3357–3381, https://doi.org/10.1016/0045-6535(95)00187-D, 1995.

**Table 1. The number of significant digits presented here seems hard to justify.**

The significant digits in Table 1 have been reduced.

**Section 6: I found myself struggling to determine how useful the analysis and discussion were in this section. The claim that this is a "new chemical regime" seems a bit overdramatic. Yes, WFM was dominated for decades by acidic sulfate conditions, but other parts of the world have long known cloud compositions that look more like the "new chemical regime" at WFM. The idea that species other than sulfate and H+ might be important contributors to cloud composition is not new.**

We are disappointed to find that we were not able to clearly convey that there indeed has been a dramatic change in the chemical regime experienced at WFM, with annual median TOC concentrations in cloud water doubling over the last decade, which was not expected and has not been explored by anyone previously, to our knowledge. We emphasize that we are referring here to absolute (not relative) abundances in TOC. This finding is one of the primary results of the paper, which we attempt to better emphasize in the revised manuscript. To what extent the trend in TOC concentrations may be linked to sulfate chemistry is unclear. Though we do expect that reduced acidity could lead to greater partitioning of organic acids from the gas phase, historical cloud water data from WFM showed similar TOC concentrations similar to today in spite of much greater acidity (shown in Fig. R1 below). The growing difference

between NH4 and SO4 + NO3 trends about halfway through the long-term monitoring period at WFM is the other major change exhibited by cloud water at WFM, which is behind our assessment of a new chemical regime and, again, is likely due to more than simply changes in sulfate chemistry. Crucially the system does not seem to be reaching a steady state TOC concentration, even as the SO4 concentrations have been leveling out.

[Figure]

Figure R1. Annual mean concentrations of TOC (right axis) and major inorganic ions (left axis) throughout the long-term monitoring program at WFM compared to concentrations published in the scientific literature for select samples collected at WFM from 1987 to 1992.

**A stronger case for the change in regime would be made by having more complete speciation information. Organic acids are likely key, as hypothesized, but not measured. Ca, Mg, Na, and K should really be measured as cations to do the analyses included here. And why do the authors largely ignore K and Na in this analysis?**

Measurements of organic acid concentrations are not available for the majority of the 28 year long-term monitoring program.  As mentioned previously, we plan to report on our recent organic acid measurements, in comparison to historical measurements conducted for a small subset of samples in 1987-1992 (some of which have been published, and others that were not, but all were conducted prior to the start of the routine measurements evaluated in this paper), but the organic acid data will require substantial additional discussion since no one to our knowledge has attempted to conduct routine measurements of organic acids in cloud water to date. Given that the current paper is focused on the routine long-term measurement program, we did not feel it would be appropriate to include that dataset and discussion here.

Based on the analysis discussed in Section 4 of the supplement and described above, we believe the measured Ca and Mg in cloud water were in their cationic form. For the past four years, especially, we are confident that the reported metals are water soluble, and therefore ionic, since we now routinely filter our cloud water samples.  As shown below, K, Na and Cl contribute a very small fraction of the total ions in the system (Fig. R2), and have even less impact on the

pH$_{TD}$ calculation (Figure R3). Figure R2 shows the overall ion balance versus the ion balance only including H$^+$, NH$_4^+$, Ca$^{2+}$, Mg$^{2+}$, SO$_4^{2-}$ and NO$_3^-$. The slope of the regression line is 0.99 and linear correlation coefficient is 0.96, indicating that the removal of K$^+$, Na$^+$ and Cl$^-$ generally had very little impact on the overall ion balance.

[Figure]

Figure R2. Comparison of Ion Balance (including all measured ions) vs. Major Ion Balance (including only H$^+$, NH$_4^+$, Ca$^{2+}$, Mg$^{2+}$, SO$_4^{2-}$ and NO$_3^-$). The difference between the two calculations is the contribution from Na$^+$, K$^+$ and Cl$^-$ ions, which are only included in the values reported on the y-axis.

[Figure]

Figure R3. Calculation of annual median pH$_{TD}$ including K, Na and Cl (dark blue) in comparison to the original calculations for pH$_{TD}$ (light blue), pH$_{BU}$ (orange) and measured bulk cloud water pH (black). Light blue curve lies underneath dark blue curve. Note that we now exclude samples with negative [H$_+$]$_{BU}$ from the annual median pH$_{BU}$ calculated here as well as in the paper (they were mistakenly included in the submitted paper, potentially yielding a high bias in the previous calculation for annual median pH$_{BU}$)

**The authors' constructs of pHTD, pHBU, and MAF in equations 6-8, are perhaps as confusing as helpful. MAF, in particular, is simply an ion balance equation, although K+, Na+, and Cl- are neglected. Section 6 goes on for many pages. At a minimum, I suggest the authors streamline the portions presented in the main text and move some of the material to supplementary information.**

We are disappointed that the reviewer did not appreciate the analysis and discussion we presented on how the bulk cloud water composition measurements could be interpreted in light of expected aerosol mixing state. While not perfect (mostly because the long-term observations were not adequately constrained to fully assess the impact from mixing state), we believe that we provided compelling evidence to support making these simple calculations to estimate the pH of the vast majority of cloud droplets as they existed in the atmosphere. The available size resolved aerosol composition measurements of $Ca^{2+}$ and $Mg^{2+}$ supported making the assumption that these analytes are present primarily within a small fraction of droplets containing coarse mode mineral dust particles. However, we did not feel we had adequate support for making that same conclusion about the other trace metal ions measured, so we did not include Na, K and Cl in the calculation of $pH_{TD}$. As WFM is an inland site, we do not experience substantial impacts from sea salt, but we also would not expect NaCl to impact the bulk cloud water pH. As we showed above, including Na, K and Cl in the $pH_{TD}$ calculations has a negligible impact on the results (See Figure R3). We posit that the substantially improved correlation between many of the measured analytes including TOC, $NH_4^+$, $SO_4^{2-}$ and $NO_3^-$ with $pH_{TD}$ suggests that the calculation for $pH_{TD}$ is a better proxy for cloud droplet pH than the bulk cloud water pH. However, we also show that $NH_4^+$, $SO_4^{2-}$ and $NO_3^-$ are increasingly unable to explain the inferred cloud droplet pH.

While the MAF calculation does account for ion balance (within a dilute aqueous system), it is not simply an ion balance calculation. See comparison of MAF to normalized ion balance in Fig. R4 below. Only at very low $NH_4^+$ concentrations are they roughly equivalent (with slight differences then due to the other trace analytes $Cl^-$, $Na^+$ and $K^+$). Though imperfect, our MAF calculation provides a quantitative estimate for how well the measured $SO_4^{2-}$, $NO_3^-$ and $NH_4^+$ concentrations can explain the estimated cloud droplet acidity. Even under the conservative estimate for MAF that only includes samples with positive values for $[H^+]_{BU}$ (i.e. with samples represented as black dots in Fig. R4 excluded), ~60% of the inferred cloud droplet acidity is unexplained by the measured $SO_4^{2-}$, $NO_3^-$ and $NH_4^+$ concentrations on an annual median basis for the last few years, which is a new and important results, and a considerably different situation from 15 years ago at WFM.

We attempt to simplify this section to more succinctly and clearly convey these points.

[Figure]

Figure R4. Missing Acid Fraction (MAF) versus Normalized Ion Imbalance for individual cloud water samples, colored by the measured $NH_4^+$ concentrations divided by the measured $SO_4^{2-}$ + $NO_3^-$ concentrations. For a normalized ion imbalance of 0.2, MAF can vary from roughly 0.2 to 1.0, corresponding to the degree to which $NH_4^+$ neutralizes $SO_4^{2-}$ and $NO_3^-$ concentrations.

**Responses to Reviewer 2**

We wish to express our sincere appreciation to the reviewer for their thorough review, which has substantially improved the paper. Our responses are as follows:

**I share the first reviewers concerns about the use of cations in the analysis and whether they were in ionic forms or not. Cations are not only from dust but are present in trace amounts from many sources as shown in the work of Reff et al. (2009) et al.**

This is a legitimate concern and we thank the reviewers for bringing this to our attention. Note that for all cloud water samples that were filtered during collection (as we have been routinely doing since taking over the cloud water monitoring program in 2018), we can be more confident that the reported measurements for calcium and magnesium are cationic, since any insoluble particles >0.4um would have been removed through filtration prior to analysis. Also note that the most recent years are the most impacted by calcium (with the greatest measured cation to anion ratio and the greatest difference between the measured bulk cloud water pH and estimated cloud droplet pH), in spite of filtration.

However, for the majority of the long-term monitoring program, samples were not filtered. The assumption has long been that all measured calcium and magnesium were fully dissolved within cloud water due to the dilute aqueous solutions that comprise cloud droplets. This assumption has been "baked in" to every previous analysis conducted with this dataset to our knowledge (including Aleksic et al., 2009; Schwab et al., 2016; Pye et al., 2020; Lee et al., 2022), given that only the "ion balanced" samples (including Ca and Mg, assumed to be in their ionic form) have been labeled as "Valid". Since this is an important assumption behind our analysis as well, we tested this assumption by conducting additional analysis of 10 unfiltered cloud water samples that were archived from 2018-2020. Only a small number of samples could be re-analyzed in this way, because we do not have archived cloud water samples prior to 2018 when routine filtering began. To our knowledge, this sort of test was never conducted on samples from the historical dataset. For our reanalysis of these samples, we used a Metrohm 761 Compact Ion Chromatography (IC) system to measure cationic $Ca^{2+}$ and $Mg^{2+}$. We re-analyzed these samples twice; once without filtering and once filtering the samples through a 0.4 μm polycarbonate filter. We found that there were virtually no differences between the filtered and unfiltered IC measurements. When compared to the original unfiltered elemental measurements, we also found very little difference (over a wide range of pH values, 4.5-6.4), consistent with the measured Ca and Mg being completely dissolved for these samples.

In the supplemental material, we also show that, if $Ca^{2+}$ and $Mg^{2+}$ are associated with $CaCO_3$ and $MgCO_3$, we expect them to be completely dissolved for the full range of pH and dilution that have been observed for cloud water samples collected at WFM. Note that other forms of calcium and magnesium, e.g. $CaCl_2$, $Ca(NO_3)_2$ or $Ca(OH)_2$, are even more soluble and even more likely to be completely dissolved at the pH and liquid water contents encountered at WFM.

Detailed information about the IC analysis and the $CaCO_3$ and $MgCO_3$ solubility calculations as a function of pH described above can be found in section 4 of the supplemental material.

Reff et al. (2009) identify calcium originating primarily from Unpaved Road Dust, Agricultural Soil, Bituminous Combustion (i.e. fly ash) and Construction Dust. These potential sources for calcium aerosol do not detract from our argument that the calcium found in cloud droplets resides almost exclusively in supermicron aerosol and not in submicron aerosol (the latter of which comprise the vast majority of particles acting as CCN). While several additional combustion sources for calcium are identified in Reff et al. (2009), atmospheric sources of calcium from e.g. Wildfires or Prescribed Burning are also likely to be supermicron aerosol associated with mineral dust lofted by updrafts rather than secondary mechanisms like condensation of calcium onto submicron aerosol, on account of the extremely high temperatures required to melt or vaporize calcium (e.g. $CaCO_3$ has a melting point of 1500-2400 Fahrenheit). However, we attempt to clarify in the manuscript that we are making an assumption about the calcium mixing state, which is supported by only a handful of measurements in the region, and that additional measurements would be needed to better constrain the cloud droplet pH estimate.

Aleksic, N., Roy, K., Sistla, G., Dukett, J., Houck, N., & Casson, P. (2009). Analysis of cloud and precipitation chemistry at Whiteface Mountain, NY. Atmospheric Environment, 43(17), 2709-2716. https://doi.org/10.1016/j.atmosenv.2009.02.053

Lee, J. Y., Peterson, P. K., Vear, L. R., Cook, R. D., Sullivan, A. P., Smith, E., et al. (2022). Wildfire smoke influence on cloud water chemical composition at Whiteface Mountain, New York. Journal of Geophysical Research: Atmospheres, 127, e2022JD037177. https://doi.org/10.1029/2022JD037177

Pye, H. O., Nenes, A., Alexander, B., Ault, A. P., Barth, M. C., Clegg, S. L., ... & Zuend, A. (2020). The acidity of atmospheric particles and clouds. Atmospheric chemistry and physics, 20(8), 4809-4888. https://doi.org/10.5194/acp-20-4809-2020

Schwab, J. J., Casson, P., Brandt, R., Husain, L., Dutkewicz, V., Wolfe, D., ... & Dukett, J. E. (2016). Atmospheric chemistry measurements at Whiteface Mountain, NY: Cloud water chemistry, precipitation chemistry, and particulate matter. Aerosol and Air Quality Research, 16(3), 841-854. https://doi.org/10.4209/aaqr.2015.05.0344

**Can more information be provided on the trends in cloud water itself? Based on the criteria on Page 5 for cloud collection, has the fraction of the year when samples are collected changed over time? This could help link changes in cloud water composition to changes in general atmospheric state. How often are cloud samples being dumped due to bottles being**

**too full? Given the role of valid/invalid samples in the analysis, some commentary on the actual sample coverage would be useful.**

The cloud water collection season is approximately from June 1$^{st}$- September 30$^{th}$ with slight variability year to year due to construction projects on the observatory or weather conditions (for example, some years start several days late or end several days early to prevent riming of the outdoor components of the cloud water collection system when inclement spring or fall weather is expected). We report the number of samples collected each year in Figure 4 of the manuscript, which is dictated mostly by sample collection rate (hourly, 3-hourly or 12-hourly).

In further consideration of this question, we now evaluate the fraction of time that the measured meteorological conditions at the summit of WFM allowed for cloud water collection within each summer month over the period 2009-2021 (as shown in Figure R1 below). The September data ends in 2020 because deployment ended early in 2021 for renovations of the silo roof. In our analysis shown below, there is some evidence that June cloud events at WFM have grown less frequent in recent years, but June also experienced the greatest year to year variability. The other months may also show very slight declines (not statistically significant). The same meteorological data for this analysis is not available prior to 2009. However, fractions of time with LWC > 0.05g m$^{-3}$ from 2001-2010 were reported by Schwab et al. (2016) (copied below in Figure R2), which show substantial variability but no obvious seasonal or long-term trends. Note that we had already shown in the supplement the median LWC over the long-term monitoring program for both valid and invalid datasets, which did not exhibit significant trends.

[Figure]

Figure R1. Percent of time that the measured meteorological conditions at the summit of WFM allowed for cloud water collection (temperature > 2C, wind speed > 2m/s, LWC > 0.05g m$^{-3}$ and no rain detected), separated by month (6 = June, 7 = July, 8 = August, 9 = September). Time periods when these measurements were not available, due to weather or instrumentation issues, are not included.

[Figure]

Figure R2. From Schwab et al. (2016)

Unfortunately, we do not have all the necessary information to report on how frequently the 1L accumulator was filled before the one-hour sampling period was over (after which the cloud collector would withdraw back into its housing, according to Baumgardner et al., 1997). This information would be relevant to all samples prior to 2015, since the 3-hourly sampling implemented 2007-2013 was accomplished by combining up to 3 one-hour samples, and the 12-hourly sampling implemented in 2014 was accomplished by combining up to 12 one-hour samples. In 2015, the 12L accumulator was installed so that cloud water never filled the accumulator after shifting to a 12-hourly collection cycle. With this current system, while extra cloud water is sent to waste once the 1-liter ISCO bottle is full, the 1L sample retained is an aliquot of the greater volume contained within the accumulator, and no sampling time is lost due to insufficient accumulator capacity. We might expect that, given cloud water tends to be more enriched in pollutants at the leading edge of the sampling period and become more dilute after several hours of collection (e.g. Khwaja et al., 1995), that samples with limited collection volume (i.e. which potentially occurred on occasion prior to 2015) would have higher concentrations than if they had had unlimited collection volume capacity. That type of bias we would expect could make the observed increasing TOC trend appear to be lower than it actually was. The potential impact on ion imbalance is unclear.

In consideration of this question, we also added another figure to the supplement (Figure S3) to show the percentage of invalid samples each year separated by month, which shows that "valid" and "invalid" samples are found in (and are growing more frequent in) all summer months. Much of the month-to-month variability in the "Percent Invalid" is due to the different thresholds used for low concentration and high concentration samples that have traditionally been applied for validity criteria based on ion balance. The cation/anion ratio, as reported in Fig. 5 of the manuscript, has shown steadier growth and is a better measure because it does not contain these arbitrary shifting thresholds.

As mentioned in the manuscript, precipitation (rain and snow) chemistry data from the base of WFM also shows a steady increase in cation/anion ratio (shown below by season in Figure R3, with the summer cloud water collection season corresponding to the green trace), which likewise correspond to a growing fraction of "invalid" samples. Given that rain water and cloud water collection typically do not take place at the same time (since rain events are intentionally excluded from the cloud water sampling periods), and the rain water samples are handled completely independent of the cloud water samples, the good agreement between cloud water and rain water observations provide evidence that neither sampling artifacts nor shifting meteorological conditions are likely to be major driving factors behind the growing trend in cation/anion ratios observed in both datasets.

[Figure]

Figure R3. Annual median cation/anion ratios for rain water samples collected at the base of WFM, reported by the National Acid Deposition Program (NADP), separated by season (FMAM = February, March, April, May; JJAS = June, July, August, September; ONDJ = October, November, December, January).

Baumgardner, R. et al. (1997), Development of an automated cloud water collection system for use in atmospheric monitoring networks. Atmospheric Environment, 31(13), 2003-2010. https://doi.org/10.1016/S1352-2310(96)00325-1

Khwaja, H. A., Brudnoy, S., & Husain, L. (1995). Chemical characterization of three summer cloud episodes at Whiteface Mountain. Chemosphere, 31(5), 3357-3381. https://doi.org/10.1016/0045-6535(95)00187-D

Minor comments:

1. **The code availability is excellent. Consider creating a persistent identifier (doi) for the github code as well. Several free services are available.**

We thank the reviewer for pointing this out. A DOI was created for the github code to help ensure reproducibility of our results, which can be found here:

https://doi.org/10.5281/zenodo.7379622

2. **Is there any information on the seasonality and likely WSOC parent hydrocarbons important for the unmeasured anions? Are they likely biogenic or anthropogenic?**

The seasonality of TOC is something we neglected to include within the manuscript. A new figure (Figure S9) was added to the supplement to remedy this oversight. June and July typically exhibit the highest concentrations of TOC while September exhibits the lowest concentrations. Theil-Sen regression analysis shows that June, July and September all have statistically significant increasing trends, with June and September driving the yearly trend. No other measured analytes in WFM cloud water exhibit consistent seasonality. Text discussing this figure was added to the manuscript.

Unfortunately, we have no measurements relevant to parent hydrocarbons that coincide with the TOC measurements at WFM. The increased concentrations of TOC in June and July suggests that perhaps there is an important biogenic component to TOC during the growing season, but that is not something we can substantiate at present given the lack of chemical speciation data. We also looked at seasonality of the TOC versus ion imbalance (Figure R4 below), which indicated higher slopes (more ion balance for a given TOC concentration) in June and September and more variability in July. Potassium concentrations in excess of 5mg/L (frequently found by Cook et al., 2017 and Lee et al., 2022 to distinguish biomass burning influenced cloud water) were frequently shown to coincide with the highest TOC concentrations and highest ion imbalance across all months, also shown in Figure R4, which could indicate that much of the ion imbalance is associated with biomass burning smoke influence.

Seasonality of potential smoke impact on WFM cloud water samples was discussed in the recently published paper Lee et al. (2022). In that study, the high smoke probability samples were more frequent in June and July than in August and September. However, unfortunately, those authors chose to only include the so-called "valid" data in their analysis, which excludes a large number of samples in recent years (and we know from our study that those samples also tend to have much higher TOC concentrations).

Cook, R.D. et al., 2017, Biogenic, urban, and wildfire influences on the molecular composition of dissolved organic compounds in cloud water, Atmospheric Chemistry and Physics, 17, 15167-15180. https://doi.org/10.5194/acp-17-15167-2017

Lee, J. Y., Peterson, P. K., Vear, L. R., Cook, R. D., Sullivan, A. P., Smith, E., et al. (2022). Wildfire smoke influence on cloud water chemical composition at Whiteface Mountain, New York. Journal of Geophysical Research: Atmospheres, 127, e2022JD037177. https://doi.org/10.1029/2022JD037177

[Figure]

Figure R4. Measured ion imbalance (Cations –Anions) versus TOC concentrations (measured in 2009-2017 and inferred from WSOC measurements in 2018-2021) for cloud water samples collected at WFM from 2009 to 2021, separated by month. Note that markers colored grey have $K^+$ concentrations exceeding the maximum 10mg/L of the colorscale.

3. **Figure 5: Could measurement uncertainty be propagated to the cation/anion ratio?**

Median standard error bars are added to the cation/anion ratio plot in Figure 5 to better capture the sample to sample variability.

4. **Is cloud water S always in the form of inorganic sulfate (SO4 2-)? Are hydroxymethanesulfonate or isoprene organosulfates ever included in the cloud water sulfate concentrations?**

Hydroxymethanesulfonate and isoprene organosulfates have never been included in the WFM cloud water monitoring measurements, so we cannot say to what degree they might have been present in cloud water.

5. **What is the likely source of WSOC and insoluble OC in this data set? Are they likely changing over time. Figure 3 plots with both WSOC and TOC trends are very helpful.**

We currently do not have the capability of determining the sources of insoluble OC nor the potential changes in insoluble OC over time in this dataset, as these observations were not made during the long-term monitoring program.  The paper mentioned previously (Lee et al, 2022) discussed insoluble residual particles, but only for 5 samples in 2014 and 2015 (not enough to evaluate trends).  Lance et al. (2020) showed evidence

that at least some black carbon aerosols were effectively wet scavenged and incorporated into cloud droplets during a 2017 pilot study at WFM.  It's possible that the concentration of insoluble particles has decreased, as the ratio of elemental to organic carbon concentrations in PM2.5 have decreased, based on measurements across NY state from 2001-2015 (Blanchard et al., 2019). However, PM2.5 and cloud water may not be experiencing the same trends in WSOC/TOC, as cloud water can be influenced by particles larger than 2.5um (as we believe to be the case with calcium and magnesium containing particles) and cloud water can also be influenced by dissolved organic gases.

Blanchard, C.L., S.L. Shaw, E.S. Edgerton, J.J. Schwab (2019). Emission influences on air pollutant concentrations in New York state: II. PM2.5 organic and elemental carbon constituents, Atmospheric Environment, 3, 100039, https://doi.org/10.1016/j.aeaoa.2019.100039

Lance, S., et al. (2020). Overview of the CPOC Pilot Study at Whiteface Mountain, NY: Cloud Processing of Organics within Clouds (CPOC). Bulletin of the American Meteorological Society, 101.10, E1820-E1841, https://doi.org/10.1175/BAMS-D-19-0022.1

6. **Could some bounding analysis be performed on the amount of organic acids needed to reconcile the data and how that fits with the measured WSOC and likely abundance of organic acids?**

We thank the reviewer for this comment as it led us to think in more detail about estimating the contribution of organic acids to TOC. By assuming the missing anions are organic acids, we can estimate the fraction of TOC that is ionic. We introduce the carbon to charge ratio (C/z), which is the ratio of carbon atoms to carboxylic acid functional groups in a given compound. We then multiply this ratio by the ion imbalance to estimate the contribution of organic acids to TOC. Four different C/z ratios are selected to give a range of potential contributions; C/z = 1 representing compounds like formic and oxalic acid, C/z = 2 representing compounds like acetic and succinic acid, C/z =3 representing compounds such as lactic and propionic acid, and C/z = 5, potentially representing large organic molecules often classified as humic-like substances. This analysis shows that organic acids could contribute as much as 12% of TOC in WFM cloud water if C/z = 1 and 58% of TOC in WFM cloud water if C/z = 5, and Theil-Sen regression indicates significant increasing trends since 2009 (p = 0.0327). A section discussing this finding was added to the manuscript and a new figure was added to the supplementary material (Figure S15).

---

## Author Response (AR2)

We would like to the thank the editor for their suggested edits.

**I. 139: replace reference by Van Stempvoort and Biggar, 2008 by (Šantl-Temkiv et al., 2022) since this reference refers to atmospherically relevant bacteria**

Šantl-Temkiv, T., Amato, P., Casamayor, E. O., Lee, P. K. H. and Pointing, S. B.: Microbial ecology of the atmosphere, FEMS Microbiol. Rev., fuac009, doi:10.1093/femsre/fuac009, 2022.

This citation was added to the manuscript.

**I. 238: replace SO4-2 by SO42-**

The subscript was corrected on line 238.

**I. 271: Is it indeed K+ and Ca2+ or more correctly K and Ca? The same question applies to Figure S8 and to the text in I. 317.**

The subscripts were corrected for K and Ca on lines 271 and 317, and in the supplemental figure.

**I. 287/8:'June, July and September' .... 'during these four summer months' - should the latter read 'three summer months'?**

We changed "these four summer months" to "any month" to state more clearly that there is no seasonality for any of the inorganic analytes

**I. 335: Strictly, there is no pKa value for CO32-. pKa is the acid dissociation constant, but CO32- is not an acid. It would be more accurate to write '... approximately the pKa value of HCO3-'. Please add also the value 6.35 here (=-log(6.37e-7)), in agreement with the value you cite for Ka after Eq-3 (the value can be then removed in I. 345)**

We corrected the sentence to say that pKa value is for  $HCO_3^-$  rather that  $CO_3^{2-}$  and included the pKa value for  $H_2CO_3$  in this section as suggested.

**I. 371: 'Assuming that the missing anions are organic acids' is not correct. Better "Assuming that the missing anions are organic anions' or '...carboxylates'**

We corrected the organic acids to carboxylates.

**I. 437: Is 'particles' missing here? ('While these aerosol can make...')**

The word particles was added to the sentence.

I. 599: The sentence 'For samples with pH>6, which are growing more common, HCO3- could be important.' Seems to contradict earlier text in line 386 'until the

**pH approaches the pKa of CO32- ,where the influence of HCO3- then decreases.' If you replace HCO3- by CO32- in the first sentence, it could be correct. – Please check if this is what you want to say here.**

We were a bit unclear how to write this section. It was intended to read that  $HCO_3^-$  becomes a more important anion as the pH increases beyond a pH of 6, but until it reaches the pKa of  $HCO_3^-$ , where the importance of  $CO_3^{2-}$  increases. We changed the sentence to read as follows:

The dissolution of CO2 is a major source of  $HCO_3^-$  by partitioning into cloud droplets and hydrolyzing to form carbonic acid (H2CO3) with an acid dissociation constant of  $4.37 \times 10^{-7}$  (pKa = 6.36). This contribution of  $HCO_3$  from CO2 increases as pH increases until it reaches its pKa of 10.3, where the relative contribution of CO32- becomes more important.